# Differential CaKAN3-CaHSF8 associations underlie distinct immune and heat responses under high temperature and high humidity conditions

Sheng Yang[1,2,3], Weiwei Cai[1,2,3,4], Ruijie Wu[1,2,3], Yu Huang[1,2,3], Qiaoling Lu[1,2,3], Hui Wang[1,2,3], Xueying Huang[1,2,3], Yapeng Zhang[1,2,3], Qing Wu[1,2,3], Xingge Cheng[1,2,3], Meiyun Wan[1,2,3], Jingang Lv[1,2,3], Qian Liu[1,2,3], Xiang Zheng[1,2,3], Shaoliang Mou[1,2,3], Deyi Guan[1,2,3] & Shuilin He ⓘ [1,2,3] ✉

High temperature and high humidity (HTHH) conditions increase plant susceptibility to a variety of diseases, including bacterial wilt in solanaceous plants. Some solanaceous plant cultivars have evolved mechanisms to activate HTHH-specific immunity to cope with bacterial wilt disease. However, the underlying mechanisms remain poorly understood. Here we find that CaKAN3 and CaHSF8 upregulate and physically interact with each other in nuclei under HTHH conditions without inoculation or early after inoculation with *R. solanacearum* in pepper. Consequently, CaKAN3 and CaHSF8 synergistically confer immunity against *R. solanacearum* via activating a subset of NLRs which initiates immune signaling upon perception of unidentified pathogen effectors. Intriguingly, when HTHH conditions are prolonged without pathogen attack or the temperature goes higher, CaHSF8 no longer interacts with CaKAN3. Instead, it directly upregulates a subset of HSP genes thus activating thermotolerance. Our findings highlight mechanisms controlling context-specific activation of high-temperature-specific pepper immunity and thermotolerance mediated by differential CaKAN3-CaHSF8 associations.

Plants are frequently exposed to pathogen attack in their natural habitats, and accumulating evidence indicates that plant-pathogen interactions can be profoundly modified by environmental conditions[1–6], with plant immunity being repressed by high temperature stress[2,7,8] and high humidity[6,9]. A common phenomenon in crop production practice is that crop diseases are more serious under conditions of high temperature and humidity than under high temperature, high humidity and normal temperature alone. This is more prominent in solanaceous crops such as pepper and tomato[10,11], which are distributed mainly in uplands in the

warm seasons of tropical and subtropical regions, where they are accompanied by various soil-borne pathogens, including *Ralstonia solanacearum*[12]. Recurring combined stresses might exert powerful constraints on plant evolution and thus shape the evolution of immunity[13,14]. In fact, plants have evolved high-temperature-high-humidity specific immunity to compensate for the immunity impaired by high-temperature-high-humidity conditions[15]. As the responses of plants to combined stresses differ from those to individual stresses[16–18], even though plant responses to high temperature and high humidity

[1]Key Laboratory of Applied Genetics of Universities in Fujian Province, Fujian Agriculture and Forestry University, Fuzhou, Fujian, China. [2]Agricultural College, Fujian Agriculture and Forestry University, Fuzhou, Fujian, PR China. [3]National Education Ministry Key Laboratory of Plant Genetic Improvement and Comprehensive Utilization, Fujian Agriculture and Forestry University, Fuzhou, Fujian, PR China. [4]College of Horticultural Sciences, Zhejiang Agriculture and Forestry University, Hangzhou, Zhejiang, PR China. ✉e-mail: shlhe201304@aliyun.com

alone have been intensively studied, the molecular details underlying the high-temperature-high-humidity specific immunity in solanaceous plants are currently poorly understood.

Accumulating evidence indicates that upon challenge with pathogens, plants employ plasma membrane-localized pattern recognition receptors (PRRs) and intracellular R proteins that are primarily nucleotide-binding and leucine-rich repeat proteins (NLRs) to perceive pathogen-associated molecular patterns (PAMPs) and effectors of pathogens, respectively. Although with distinct amplitudes and dynamics, the immune signaling initiated by distinct perceptions is transmitted by different signaling cascades, such as phytohormones, including SA[19], JA[20], and cytokinins[21,] and accumulates in the nuclei with massive transcriptional reprogramming crucial for the activation of pepper immunity[22–25]. Transcription factors (TFs) have been implicated and act as crucial players in the regulation of transcriptional reprogramming, with hundreds or thousands of genes being governed directly or indirectly by a single TF[26–28]. Members of TF families, such as WRKY, AP2/ERF, bHLH, NAC, and TGA/bZIP, appear to be prominent regulators of host defense[29], and the molecular details of some of these TFs have been well documented[29–31]. The mechanisms underlying plant immunity mediated by these TFs appear to be complicated; for example, to activate the immune response appropriately, some of the TFs that are involved in regulating several seemingly disparate processes should fulfil their functions specifically, different TFs should work in concert and coordinately, and some TFs form positive or negative feedback loops with their target genes[32–36]. Accumulating evidence suggests that high temperature or high humidity can compromise plant immunity not only by inhibiting SA, JA, or cytokinin signaling[4,15] but also by repressing NLR proteins such as SNC1, RPW8.1, and RPW8.2[7,37–40] and TFs such as CAMTA3, PIF4, CBP60g, and SARD1[41–43], indicating that the modification of plant immunity by high temperature or high humidity might occur at multiple levels, including pathogen effector perception and the transcriptional level. However, the NLR proteins and TFs involved in high-temperature-high-humidity specific plant immunity and how they are functionally related remain to be elucidated.

In the present study, to elucidate the mechanism underlying pepper immunity against RSI under HTHH, the roles of two novel TFs and their interaction in the response of pepper plants to RSI under HTHH were functionally characterized. The first TF is CaKAN3, which belongs to the KANADI (KAN) family of proteins. KAN proteins contain GARP (GOLDEN2, Arabidopsis response regulator (ARR) and phosphorus stress response 1 (PSR1)) DNA-binding domains and EAR motif repressor-like domains (PDLSL and LEFTL) and constitute a subclade of the GARP family of MYB-like transcription factors[44]. The EAR motif repressor-like domains in KAN TFs are likely to facilitate their interactions with TOPLESS corepressors[45–48]; thus, KAN TFs act as transcriptional repressors by targeting cis-elements such as GAATA(A/T), (A/C)CAAAA, and CAAGT(T/G)G in their target genes[49,50]. Thus far, it seems that KAN TFs have been implicated exclusively in the regulation of plant growth and development, such as organ development and polarity establishment[44,51–55], the generation of gametangiophores and sporophytes[56,57], and the promotion of abaxial fate in lateral organs[55,58,59]. The second TF we functionally evaluated in the present study is CaHSF8, a member of the heat shock factors (Hsfs), which have been found to participate mainly in the regulation of the plant response to heat stress by recognizing the heat stress elements (HSE: AGAAnnTTCT) conserved in promoters of heat shock-inducible genes[60,61], in the regulation of processes such as defense signaling against other abiotic stresses such as salt and drought[62,63], and in other processes such as pollen and seed development[64,65], flowering[66], plant growth[67] and secondary metabolism[62]. In addition, HSFs have also been found to play roles in plant immunity against pathogens[35,67,68] as well as in the regulation of trade-offs between growth and defense responses[67]. However, whether and how HSFs are involved in plant immunity under HTHH is unclear. Our data demonstrated that CaKAN3 was upregulated by HTHH or RSHT, and it appeared to be involved in neither the regulation of plant organ development nor immunity against RSI at room temperature; instead, it acted positively in pepper immunity against RSI under HTHH by specifically targeting a subset of NLR genes in association with CaHSF8. These results indicate that pepper immunity against RSI under HTHH is distinct from that under room temperature, highlighting a novel context-specific transcriptional regulation of NLR genes by the CaKAN3-CaHSF8 module during the pepper response to RSI under HTHH.

## Results

### *CaKAN3* is upregulated by HTHH or RSHT

In a dataset of RNA-seq using pepper roots challenged with RSI at 37 °C and 90% humidity, a gene encoding a putative KAN3 attracted our attention. Its deduced amino acid sequence contains a conserved GARP domain but does not contain the DLSL domain and appears to be structurally conserved in the *Capsicum* genus, including *Capsicum annuum*, *Capsicum baccatum* and *Capsicum chinense* (Supplementary Fig. 1a, 1b), and we named it *CaKAN3*. In addition, several cis-elements, including AT-rich, W-box, ATCT, GARE- and G/C motifs, were found within the promoter region of *CaKAN3* (Supplementary Fig. 2a). We assayed its transcription by RT–qPCR and found that it was not upregulated by RSI at 28 °C and 90% humidity but was upregulated by both 37 °C and 90% humidity and *Ralstonia solanacearum* infection under HTHH (37 °C and 90% humidity) at 6 and 12 hpt, respectively (Supplementary Fig. 2b). By agroinfiltration-based transient overexpression, CaKAN3-GFP was observed in the nuclei of epidermal cells in the agroinfiltrated leaves of *Nicotiana benthamiana* plants (Supplementary Fig. 3).

### CaKAN3 acts as a positive regulator specifically in the pepper response to RSHT

The upregulation of CaKAN3 during the pepper response to RSI at 37 °C and 90% humidity implied its role in pepper immunity against RSHT. This speculation was supported by the data from *CaKAN3* silencing by virus-induced gene silencing (VIGS) in pepper plants, in which *CaKAN3* was silenced (Fig. 1a). Consistent with previous studies, the HTHH condition suppressed plant resistance to bacterial wilt disease (Fig. 1b, c). Interestingly, however, CaKAN3 silencing further increased plant susceptibility to bacterial wilt under the HTHH condition but not under the RSRT condition (Fig. 1d), lower levels of dynamic ROS (reactive oxygen species) accumulation upon flg22 (flagelin 22) treatment at 37 °C and 90% humidity compared to the wild-type plants (Fig. 1e), and reduced transcript levels of *CaMgst3* and *CaPRP1* (Fig. 1f), which are positively related to pepper immunity against RSI at 37 °C and 90% humidity[15]. However, the silencing of *CaKAN3* did not affect either pepper immunity against RSI at room temperature or tolerance to HTHH (37 °C, 90% humidity) (Fig. 1b–f). These data collectively indicate that CaKAN3 acts positively in the pepper response to RSI at 37 °C and 90% humidity but not at 28 °C and 90% humidity.

To confirm the result from CaKAN3 silencing, we generated CaKAN3-GFP-overexpressing *Nicotiana benthamiana* T3 lines. Two lines (#1 and #2) with higher levels of *CaKAN3* expression compared to the wild-type plants (Supplementary Fig. 4a) were selected for assay with the plants being challenged by RSI at 28 °C, 90% humidity and 37 °C, 90% humidity or with the plants challenged by the condition of 37 °C and 90% humidity alone. The results showed that, compared to the wild-type plants, CaKAN3-overexpressing lines showed enhanced resistance to RSI at 37 °C and 90% humidity but not at 28 °C and 90% humidity (Supplementary Fig. 4b), manifested by the lower level of *R. solanacearum* growth (Supplementary Fig. 4c), and the HTHH treatment alone did not affect the phenotypes of all of the tested plants (Supplementary Fig. 4b), indicating that susceptibility of wild-type plants to RSI at 37 °C and 90% humidity was not due to the

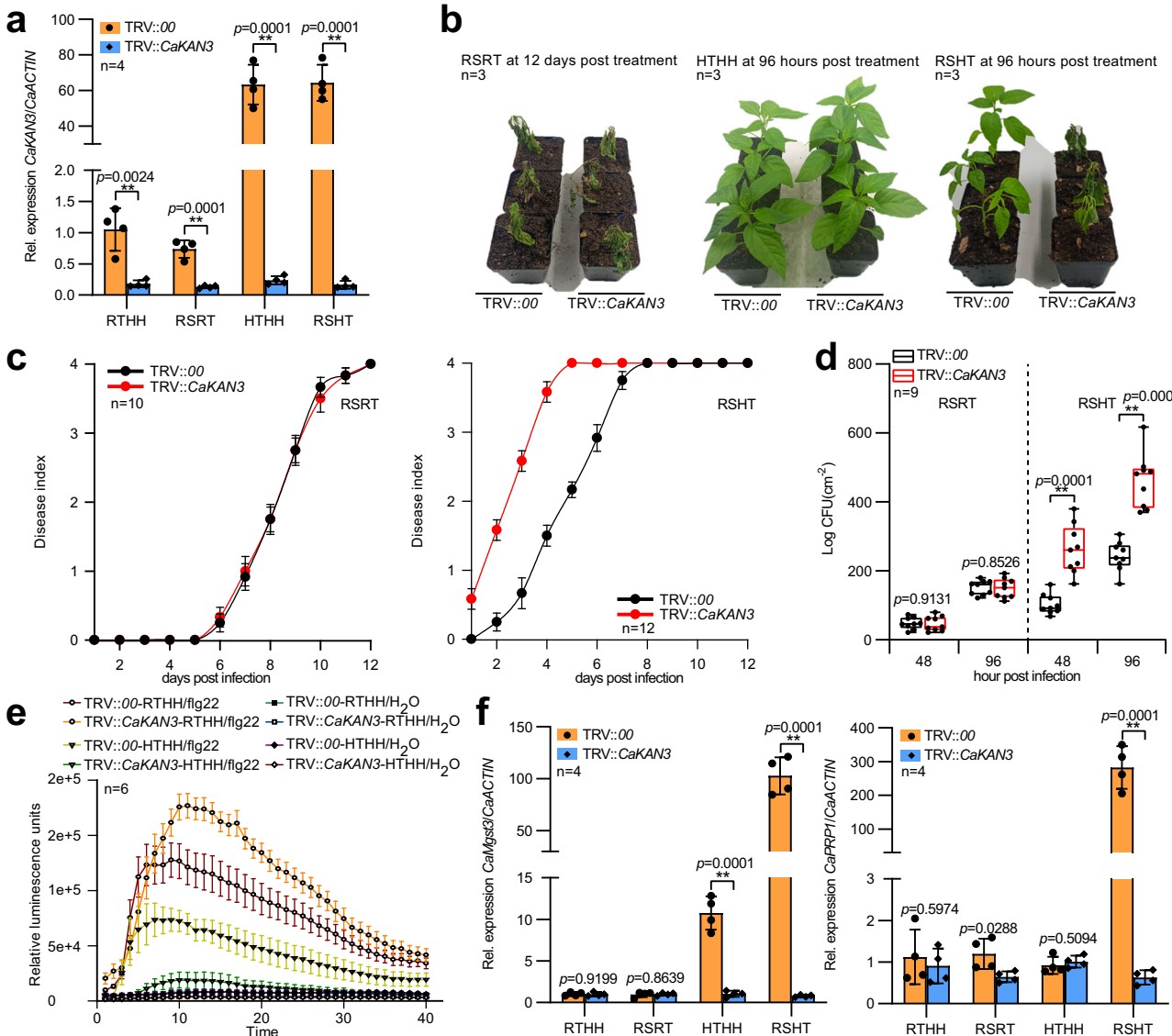

**Fig. 1 | *CaKAN3* silencing significantly increased the susceptibility of pepper plants to *R. solanacearum* inoculation (RSI) under HTHH. a** The success of *CaKAN3* silencing by virus-induced gene silencing (VIGS) by measuring the transcript levels of *CaKAN3* in RTHH-, RSRT-, HTHH- and RSHT-challenged TRV:*CaKAN3* pepper plants at 24 hours post-treatment (hpt). The transcript levels of TRV:*OO*/ RTHH were set to 1. **b** Effect of *CaKAN3* silencing on the response of pepper plants to RSRT and RSHT treatment at 3 and 12 dpt, respectively. **c** The disease index of *CaKAN3*-silenced pepper plants challenged with RSRT or RSHT from 0 to 12 dpt. A total of 24 plants were dynamically scored for each species. In **b**, **c**, the experiment was carried out three times with similar results. **d** Growth of *R. solanacearum* in *R. solanacearum*-inoculated *CaKAN3*-silenced plants at room temperature or under HTHH, shown as colony-forming units (cfu). Data are shown as the means ± standard errors of eight replicates. Asterisks above the bars indicate significant

differences between means (*P* < 0.01), as calculated with a t test. The center line represents the median value and the boundaries indicate the 25th percentile (upper) and the 75th percentile (lower). Whiskers extend to the largest and smallest value. **e** Decreased flg22-induced $H_2O_2$ production in *CaKAN3*-silenced pepper plants at HTHH. The results shown are representative of two independent experiments. Data are shown as the means ± standard errors of six replicates. **f,** Relative transcript levels of *CaMgst3* and CaPRP1 in TRV:00 and TRV:*CaKAN3* pepper plants challenged with RTHH, RSRT, HTHH or RSHT. The transcript levels of TRV:*OO*/ RTHH were set to 1. In **a** and **f**, data represent the mean ± SD of four replicates. *CaActin* was used as an internal control, and asterisks above the bars indicate significant differences between means (*P* < 0.01), as calculated with Fisher's protected t test. All replicates were from different plants. In **a–f**, source data are provided as a Source Data file.

treatment of HTHH but was due to the combination of RSI and HTHH. It can be concluded that CaKAN3 acts positively and specifically in pepper immunity against RSI under HTHH.

## CaKAN3 interacts with CaHSF8

The specific role of CaKAN3 in pepper immunity against RSHT indicates its possible context-specific activation when pepper plants are challenged by RSHT. To isolate the possible regulatory proteins, we performed an IP-MS (immunoprecipitation-mass spectrometry) assay using CaKAN3-GFP isolated from transiently overexpressing pepper

leaves challenged with RSI at 28 °C and 90% humidity or 37 °C and 90% humidity (Fig. 2a) among 10 potential interacting proteins of CaKAN3 under RSI at 37 °C and 90% humidity with high confidence (Fig. 2b). CaHSF8 was selected by primary confirmation via BiFC assay (Fig. 2c, Supplementary Fig. 5), which was further confirmed by MST (microscale thermophoresis) assay using prokaryotically expressed CaKAN3-GST and CaHSF-6×His (Fig. 2d), pull-down assay using prokaryotically expressed CaKAN3-6×His and CaHSF8-GST (Fig. 2e), and CoIP (coimmunoprecipitation) using CaHSF8-GFP or CaKAN3-Myc proteins with CaHSFA1 and CaKAN4 as negative controls, respectively (Fig. 2f).

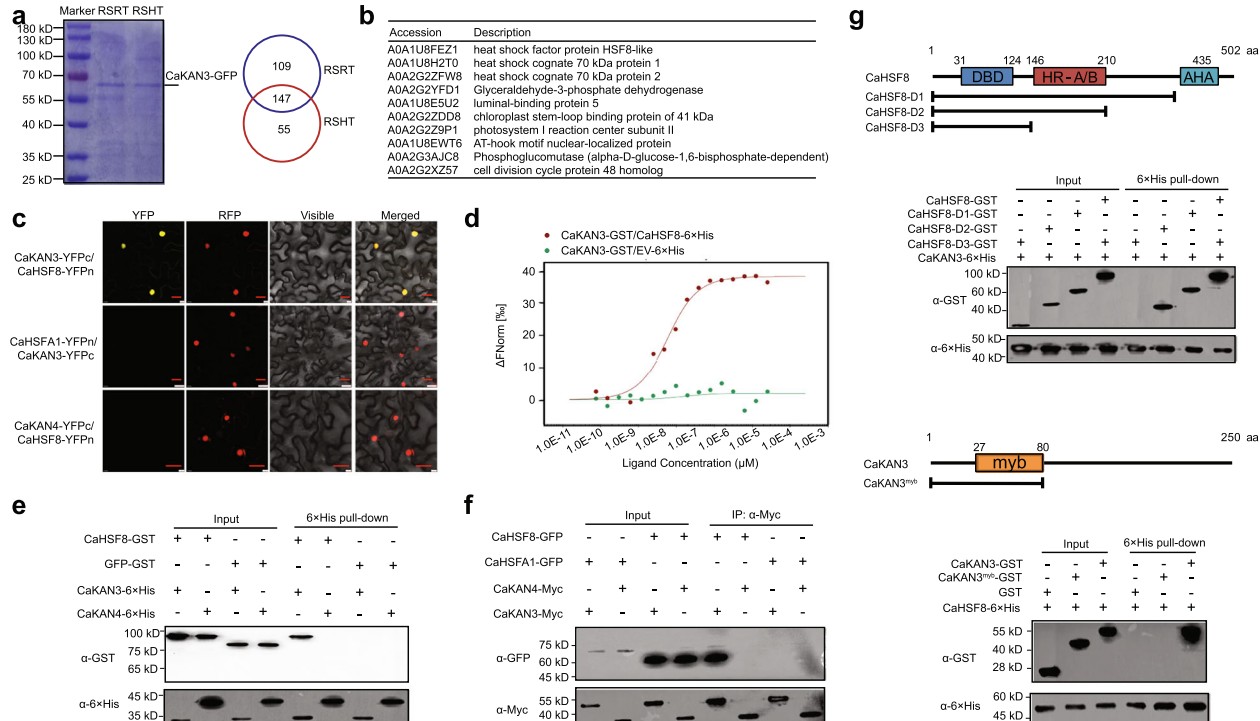

**Fig. 2 | CaKAN3 interacted with CaHSF8. a** Comparative assay of interacting proteins of CaKAN3 in pepper plants challenged by RSRT or RSHT, shown as Venn diagrams, the experiment was carried out once. **b** The top 10 specific interacting proteins of CaKAN3 with high levels of confidence in pepper plants challenged by RSHT. **c** BiFC analysis of the interaction between CaKAN3 and CaHSF8 in *N. benthamiana* leaves. NbH2B (histone H2B)-RFP was used to indicate the nucleus. Red fluorescence and yellow fluorescence, visible light and merged images were taken on a confocal microscope at 48 hpi. Bars = 25 μm. **d** In vitro interaction between CaHSF8 and CaKAN3 revealed using MST. CaKAN3-GST was regarded as the target, and the CaHSF8-6×His protein was used as the ligand and diluted to a range of concentrations from 1.0E-10 mM to 1.0E-3 mM. The mixtures of CaKAN3-GST/EV-6×His or CaKAN3-GST/CaHSF8-6×His were loaded into Monolith NT.115 capillaries, which were measured using 50% IR laser power and an LED excitation source with λ = 470 nm at ambient temperature. **e** Pull-down assay revealing the in vitro interaction between CaHSF8 and CaKAN3. CaHSF8-GST was incubated with CaKAN3-6×His and Ni Smart beads for 3 h at 4 °C under slow rotation. The bound proteins were eluted from the beads and detected using an anti-His or an anti-GST antibody. **f** Interaction between CaKAN3 and CaHSF8 in vivo, as determined by a coimmunoprecipitation assay. Proteins were isolated from pepper leaves transiently overexpressing CaHSF8-GFP/CaKAN3-Myc, CaHSFA1-GFP/CaKAN3-Myc, CaHSF8-GFP/CaKAN4-Myc and CaHSFA1-GFP/CaKAN4-Myc, which were immunoprecipitated with an anti-Myc antibody. The presence of the tested interacting proteins was detected using an antibody against GFP by western blotting. **g** Analysis of the domain of the interaction between CaKAN3 and CaHSF8 by pull-down, revealing that HR-A/B in CaHSF8 is responsible for the CaKAN3 and CaHSF8 interaction. In **c**, **f** CaHSFA1 and CaKAN4 were used as negative controls, respectively. In **c** to **g**, the experiment was carried out twice with similar results.

All the data indicate that CaKAN3 interacted with CaHSF8, and by the data from BiFC, this interaction occurred in the nuclei. We further assayed the possible domains responsible for the interaction by a pull-down assay using three CaHSF8 deletions and one CaKAN3 deletion. The results showed that D3 did not interact with CaKAN3 and that CaKAN3[myb] did not interact with CaHSF8 (Fig. 2g), indicating that the HR-A/B domain in CaHSF8 and full-length CaKAN3 are required for the CaHSF8/CaKAN3 interaction.

## CaHSF8 acts positively in pepper immunity against RSHT

To test whether CaHSF8 is involved in pepper immunity against RSRT and RSHT and in tolerance to HTHH, its transcription was assayed in pepper plants. The results showed that CaHSF8 was upregulated by HTHH (37 °C, 90% humidity) and by RSI at 37 °C, 90% humidity but not by RSI at 28 °C, 90% humidity (Supplementary Fig. 6a). By agroinfiltration-based transient overexpression, CaHSF8-GFP was found to be exclusively located in the nuclei of epidermal cells in the leaves of *Nicotiana benthamiana* plants, similar to CaKAN3 (Supplementary Fig. 6b). In addition, we studied the function of CaHSF8 by loss of function via VIGS (Supplementary Fig. 7a) and gain of function via its overexpression in *Nicotiana benthamiana* plants (T3 lines) (Supplementary Fig. 8a, b) and found that, compared to the wild-type plants, the silencing of *CaHSF8* did not affect the phenotype of plants or their tolerance to HTHH (37 °C, 90% humidity) or pepper resistance to RSI at

28 °C, 90% humidity but significantly enhanced the susceptibility of pepper plants to RSI at 37 °C, 90% humidity (Supplementary Fig. 7b). The enhanced susceptibility of pepper plants to RSI at 37 °C and 90% humidity was also displayed by an enhanced dynamic disease index from 2 to 12 dpt (Supplementary Fig. 7c), enhanced growth of infiltrated *R. solanacearum* (Supplementary Fig. 7d), and lower levels of dynamic ROS accumulation upon flg22 treatment at 37 °C and 90% humidity compared to the wild-type plants (Supplementary Fig. 7e). In contrast, we used two T3 lines (#1 and #2) of CaHSF8-overexpressing *Nicotiana benthamiana* plants (Supplementary Fig. 8a-b), which did not exhibit any phenotypic difference compared to the wild plants, and found that overexpression of CaHSF8 significantly reduced the susceptibility of *Nicotiana benthamiana* plants to RSI at 37 °C and 90% humidity (Supplementary Fig. 8c), which was displayed by enhanced growth of infiltrated *R. solanacearum* (Supplementary Fig. 8d) but did not alter the susceptibility of *Nicotiana benthamiana* plants against RSI at 28 °C and 90% humidity or tolerance to HTHH (37 °C and 90% humidity) treatment (Supplementary Fig. 8c). All these data support the result that CaHSF8 acts specifically in the pepper response to RSHT.

## CaKAN3 and CaHSF8 act positively in resistance to RSHT in different pepper inbred lines

To further study the role of CaKAN3 and CaHSF8 in pepper resistance to RSI at 37 °C and 90% humidity, their expression upon RSI at 37 °C

and 90% humidity and their roles in the RSHT response in pepper inbred lines with different levels of RSHT resistance[15] were assayed. The results showed that *CaKAN3* exhibited higher levels of transcripts upon HTHH (37 °C and 90% humidity) or RSI under HTHH in pepper lines (101-1-c-2-3, 192-3, and GZN-13-42) with higher levels of RSHT resistance than in lines with lower levels of RSHT resistance. Although *CaHSF8* was upregulated by both HTHH and RSHT in the majority of pepper lines, no obvious difference in its transcript levels among the tested pepper lines was found (Supplementary Fig. 9a). In addition, the silencing of *CaKAN3* or *CaHSF8* significantly increased RSHT susceptibility in pepper lines 101-1-c-2-3 and 192-3 (with higher levels of RSHT resistance), displayed by an increased dynamic disease index from 0 to 12 dpt and bacterial growth at 48 hpi, but did not significantly affect RSHT susceptibility in GZN13-36 and 203 (pepper line with lower levels of RSHT) (Supplementary Fig. 9b to e). In contrast, the transient overexpression of *CaKAN3* by agroinfiltration repressed bacterial growth, induced HR cell death and increased RSHT resistance in both GZN13-36 and GZN203; however, the transient overexpression of CaHSF8 did not significantly affect the pepper RSHT response in either GZN13-36 or GZN203 (Supplementary Fig. 9f to h). These data indicate that both CaKAN3 and CaHSF8 act positively in pepper resistance to RSHT, and the role of CaHSF8 as a positive regulator in pepper resistance to RSHT is CaKAN3 dependent. Furthermore, we also assayed the roles of CaKAN3 and CaHSF8 in the pepper response to *Ralstonia solanacearum* strain FJ1470 and GMI1000 inoculation (different *R. solanacearum* strains) and to *Pst DC3000* inoculation. The results showed that the silencing of *CaKAN3* or *CaHSF8* significantly increased the susceptibility of pepper plants to inoculation with both GMI1000 and FJ1470 at 37 °C and 90% humidity (Supplementary Fig. 10a), as displayed by the dynamic disease index and bacterial growth. Importantly, the data also showed that the silencing of CaKAN3 or CaHSF8 increased pepper susceptibility to inoculation with *Pst DC3000* at 37 °C and 90% humidity (Supplementary Fig. 10b, c). All these data indicate that CaKAN3 and CaHSF8 act positively in pepper response to inoculation of different *R. solanacearum* strains and *Pst DC3000* under HTHH.

## The DNA-binding sites of CaKAN3 and CaHSF8 are determined by ChIP-seq

To study the possible mode of action, the DNA sites of both CaKAN3 and CaHSF8 were determined by ChIP-seq using chromatin isolated from CaKAN3- or CaHSF8-overexpressing leaves of pepper plants challenged with RSI at 37 °C and 90% humidity, which were subjected to ChIP-seq. The results revealed 10661 and 36473 high-confidence CaKAN3 and CaHSF8 binding sites, which were associated with 717 and 2337 genes, respectively (p value < 0.05). Over 89% of the identified peak regions of CaKAN3 and CaHSF8 were distal intergenic regions, and only approximately 7–8% of those were located in promoter regions of the target genes of CaKAN3 and CaHSF8 (with 2.08% in the 1–2 kb promoter, 3.96% in the 1 kb promoter, and 1.92% in the 2–3 kb promoter) (Fig. 3a, b). Importantly, by comparative assay, 199 overlapping peak regions were found in CaKAN3 and CaHSF8 binding sites, which were enriched in pathways including plant–pathogen interactions (Fig. 3c).

Using DREME/MEME software, we determined the conserved consensus sequence AACAA within high-confidence CaKAN3 binding sites and TTCTAGAA (HSE) within high-confidence CaHSF8 binding sites across the genome (Supplementary Fig. 11a). The possible binding of these cis-elements by the two TFs was tested by an electrophoretic mobility shift assay (EMSA) using CaKAN3-GST and 3×AACAA and its mutated version 3×AAAAA. The results showed that the AACAA sequence produced a clear mobility shift, but the mutated version did not (Supplementary Fig. 11b). In addition, the activity of GUS driven by 3×AACAA plus the core 35 S core promoter was found to be enhanced by transient overexpression of CaKAN3-HA at 37 °C and 90% humidity

but weakened at 28 °C and 90% humidity (Supplementary Fig. 11c). Similarly, we found that CaHSF8 directly regulates its target genes by binding TTCTAGAA (Supplementary Fig. 11b), but the activity of GUS driven by 3×HSE plus the 35 S core promoter was found to be enhanced by transient overexpression of CaHSF8-HA at 28 °C, 90% humidity or 37 °C, 90% humidity (Supplementary Fig. 11c). These results indicate that CaKAN3 and CaHSF8 fulfil their functions by directly regulating their target genes via AACAA and HSE, respectively. Both CaKAN3 and CaHSF8 might act as TFs, but unexpectedly, by yeast one-hybrid system, we found that CaHSF8 and the positive control CabZIP63[69] all exhibited transcriptional activity, but we did not find any transcriptional activity in CaKAN3 (Supplementary Fig. 11d).

## CaKAN3 and CaHSF8 confer pepper resistance to RSHT by directly upregulating a subset of NLR genes

Among the target genes that are associated with plant–pathogen interactions by KEGG analysis, a total of six NLR genes, CaRIB23, CaRIB11, CaR1A, CaR1B12, CaR1A-6, and CaR1B12, were found to be targeted by both CaKAN3 and CaHSF8, and AACAA and HSE, which are responsible for CaKAN3 and CaHSF8 binding, were found to be present in the promoters of these genes (Fig. 3d). To test this possibility, we performed a ChIP–qPCR assay and found that all of the NLR genes were targeted by both CaKAN3 and CaHSF8, while *CaWRKY40* and *CaWRKY58*, used as negative controls, were not bound by either CaKAN3 or CaHSF8 (Fig. 3e). We found by ChIP–qPCR that there were multiple cis-elements present for CaKAN3 to potentially bind, but only the cis-element close to the HSE exhibited the highest enrichment of CaKAN3 (Supplementary Fig. 12). In parallel, the binding of these NLR genes by CaKAN3 and CaHSF8 was also tested by EMSA using their AACAA- or HSE-containing promoter fragments and their corresponding mutated versions. The results showed that CaKAN3 and CaHSF8 bound the probes (displayed by mobility shift) via the AACAA or HSE cis-element, while the mutated probes did not produce any mobility shift (Fig. 3f). In addition, although *CaMgst3* and *CaPRP1* were positively regulated by both CaKAN3-CaHSF8 modules, ChIP-qPCR showed that these two genes were not directly targeted by either CaKAN3 or CaHSF8 (Supplementary Fig. 13a, b). All these data indicate that the tested NLR genes were both targeted specifically by CaKAN3 and CaHSF8.

Since the ectopic overexpression of CaKAN3 or CaHSF8 enhanced the resistance of *Nicotiana benthamiana* plants to RSHT, to test whether this enhanced RSHT resistance is attributed to the upregulation of similar NLR genes, the homologues of the six pepper NLR genes in the *Nicotiana benthamiana* genome were searched, and a total of five homologues in the *Nicotiana benthamiana* genome were found (Supplementary Fig. 14a). The possible CaKAN3- or CaHSF8-responsive cis-elements within their promoters were assayed (Supplementary Fig. 14b). Some of these cis-elements were found to be bound by CaKAN3 or CaHSF8 by ChIP-PCR (Supplementary Fig. 14c) and further by ChIP-qPCR using specific primer pairs (Supplementary Fig. 14d). Consistently, the five NLR genes were found to be significantly higher in HTHH (37 °C, 90% humidity)-challenged *Nicotiana benthamiana* plants at 1 hpt than in the wild-type control plants (Supplementary Fig. 14e). The data indicate that CaKAN3 or CaHSF8 overexpression conferred RSHT resistance by activating NLR genes.

## Six NLRs are upregulated by HTHH only at the early stage and act positively in pepper immunity against RSHT

To further confirm the role of CaHSF8 and CaKAN3 and their interaction in pepper immunity against RSI related to HTHH, we studied the expression profiles of the 6 NLR genes upon HTHH (37 °C, 90% humidity) treatment. We first studied their transcription in HTHH-challenged pepper plants at different time points by common transcriptome database (Pepperhub: http://pepperhub.hzau.edu.cn/) and found that these genes were upregulated at the early stage (1 to 3 hpt)

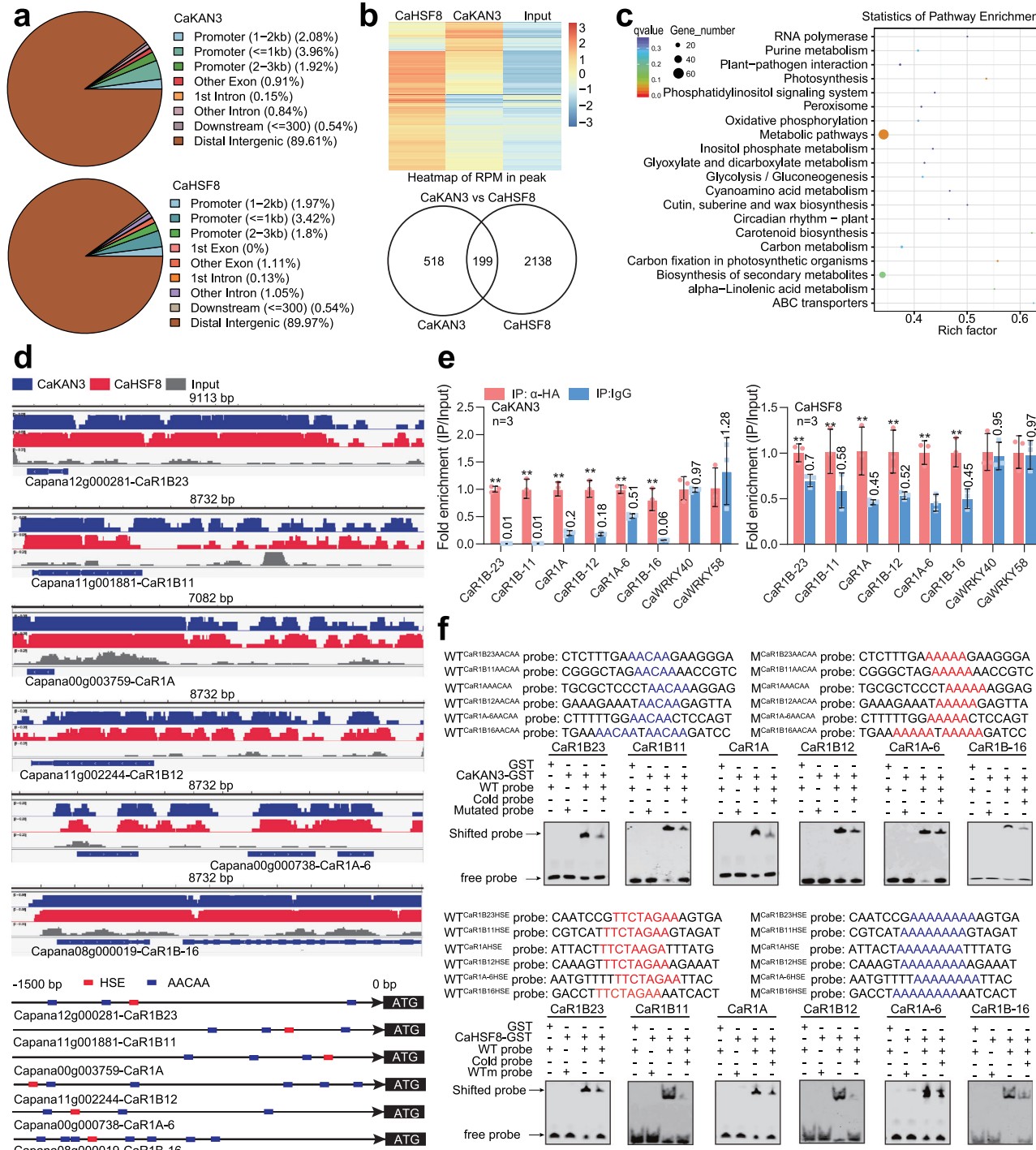

**Fig. 3 | Determination of DNA-binding sites and target genes of CaKAN3/CaHSF8 by ChIP-seq. a** Genome-wide distribution of DNA-binding peaks of CaKAN3/CaHSF8. **b** The reads per million (RPM) of CaHSF8, CaKAN3 and input are shown as a heatmap, and 199 cotargeted genes by CaHSF8/CaKAN3 are shown as Venn diagrams. **c** The genes cotargeted by CaHSF8/CaKAN3 were enriched in different KEGG signaling pathways. **d** Integrative Genomics Viewer (IGV) images of ChIP-seq data and the locations of HSE and AACAA motifs within the promoters of the *CaR1B23, CaR1B11, CaR1A, CaR1B12, CaR1A6* and *CaR1B-16* genes that are cotargeted by CaHSF8/CaKAN3. **e** Both CaKAN3 and CaHSF8 exhibited higher levels of enrichment on the promoters of the 6 tested NLR genes. GV3101 cells containing *35 S:CaKAN3-HA* and *35 S:CaHSF8-HA* were infiltrated into pepper leaves, which were harvested at 48 hpi for ChIP–qPCR analysis using specific primer pairs.

IP using IgG beads was used as the control. The enrichment levels of the tested genes were compared with those in the control, and the relative enrichment of IP using anti-HA was set to a value of 1 after normalization by input. Data are shown as the means ± standard errors of three replicates. Asterisks above the bars indicate significant differences between means ($P < 0.01$), as calculated with Fisher's protected t test. CaWRKY40 and CaWRKY58 were used as negative controls. The ratio of IP:anti-HA to IP:IgG is indicated on the error line of IP:IgG. All replicates were from different plants. Source data are provided as a Source Data file. **f** CaKAN3-GST and CaHSF8-GST bound the promoters of the 6 NLR genes in an AACAA-element- and HSE-dependent manner, as shown by EMSA. The experiment was carried out twice with similar results.

of HTHH treatment and downregulated thereafter (Supplementary Fig. 15a). These results were confirmed by measuring the activity of GUS driven by native promoters of the 6 NLR genes based on agroinfiltration in pepper leaves and by LUC assay, and we found that the expression of GUS driven by these promoters occurred only at the early stage (1 hpt) of HTHH treatment (Supplementary Fig. 15b, c). We also tested the expression profiles of the 6 NLR genes at different temperatures from 28 °C to 45 °C and found that the 6 NLR genes were upregulated by temperatures from 31 °C to 37 °C (Supplementary Fig. 15d), and these results were further confirmed by LUC assays (Supplementary Fig. 15e). To assay the functions of the 6 NLR genes in pepper immunity against RSHT, we successfully silenced these genes individually using their specific 3′UTR 300-500 fragments for vector construction (Supplementary Fig. 16a), and their silenced pepper plants were challenged by RSI at 28 °C, 90% humidity, 37 °C, 90% humidity or 42 °C, 90% humidity and HTHH (37 °C, 90% humidity) treatment alone. The silencing of these NLR genes individually only increased pepper susceptibility to RSI at 37 °C and 90% humidity (Supplementary Fig. 16b), displayed by a high dynamic disease index (Supplementary Fig. 16c) and a higher level of bacterial growth (Supplementary Fig. 16d) as well as downregulation of *CaMgst3* and *CaPRP1* related to immunity against RSHT (Supplementary Fig. 16e). Among the five *Nicotiana benthamiana* NLR genes, the silencing of *NbR1B23*, *NbR1B11* or *NbR1B12* significantly increased the susceptibility of *Nicotiana benthamiana* plants to RSI at 37 °C and 90% humidity (Supplementary Fig. 17a to d). The data indicate that the 6 pepper NLR genes all act as positive regulators of pepper immunity against RSHT.

### CaR1B-11 interacts with CaR1B-23, CaR1B-12, CaR1A, CaR1A-6, and CaR1B-16 in vivo or in vitro, probably forming a complex

As the six pepper NLR genes were collectively targeted and upregulated by the CaKAN3-CaHSF8 module, we speculate that these NLR proteins might be functionally related to each other in some way. To test this speculation, we assayed the possible interaction among the six NLR proteins first by BiFC and found that CaR1B-11 interacted with the other five NLR proteins in vivo (Supplementary Data Fig. 16f). This result was further confirmed by an MST assay using prokaryotically expressed proteins with fluorescently labeled CaR1B-11, and the results showed that CaR1B-11 interacted with the other five NLR proteins in vitro (Supplementary Data Fig. 16g). These results indicate that the six NLR proteins might be functionally related by forming a complex.

### Interaction between CaHSF8 and CaKAN3 in the regulation of the six NLR genes

To study the effect of the CaHSF8 and CaKAN3 interaction on the 6 NLR genes, we studied the effect of silencing one gene on activating the 6 NLR genes by overexpression of the other. The results showed that upon HTHH (37 °C, 90% humidity), all of the tested NLR genes were activated by CaHSF8 transient overexpression at 0 and 1 hpt, downregulated at 0 hpt by transient overexpression of CaKAN3 but upregulated at 1 hpt. Consistently, the upregulation of the tested NLR genes by CaHSF8 transient overexpression was blocked by silencing *CaKAN3* at 1 hpt but not at 0 hpt, and when *CaHSF8* was silenced, the 6 tested NLR genes were negatively regulated by transient overexpression of CaKAN3 (Fig. 4a, b, Supplementary Fig. 18). We also studied the effects of the co-transient overexpression of CaKAN3 and CaHSF8 on the expression of the 6 NLR genes. The results showed that the co-transient overexpression of CaKAN3 and CaHSF8 increased the transcript levels of the 6 NLR genes compared with the transient overexpression of CaHSF8 alone, although they were downregulated by transient overexpression of CaKAN3 (Fig. 4c). In addition, we tested the binding of CaKAN3 and CaHSF8 to the promoters of the 6 NLR genes and the effect of their interaction by ChIP–qPCR. The results showed that the binding of CaKAN3 to the promoters of the 6 NLR genes was not affected by *CaHSF8* silencing, but the binding of CaHSF8

to these promoters was significantly reduced by *CaKAN3* silencing (Fig. 4d). These results indicate that CaKAN3 is required for the targeting of CaHSF8, while CaHSF8 is required for the transformation of CaKAN3 from a negative regulator to a positive regulator in the transcription of NLR genes during the pepper response to RSHT.

### CaHSF8 acts positively in pepper tolerance to extreme temperature and high humidity

Although *CaHSF8* silencing did not result in phenotypic damage upon HTHH (37 °C, 90% humidity) at 96 hpi (Supplementary Fig. 7b), it did cause alteration in plant thermotolerance at 42 °C since the overexpression of CaHSF8 significantly enhanced *Nicotiana benthamiana* thermotolerance, as indicated by a higher survival rate from 8 to 14 dpt, enhanced maximal effective quantum yield of photosystem II (Fv/Fm) and actual photochemical efficiency of PSII in the light (Φ PSII), and a lower level of $H_2O_2$ accumulation displayed by DAB staining, which are all closely related to plant thermotolerance[34]. In contrast, the overexpression of CaKAN3 did not affect *Nicotiana benthamiana* thermotolerance (Supplementary Fig. 19a–d, Supplementary Fig. 20c, d). Consistently, *CaHSF8* silencing significantly reduced pepper thermotolerance, as displayed by reduced values of Fv/Fm and Φ PSII, lower levels of DAB staining and a reduced plant survival rate from 4 to 12 dpt (Supplementary Fig. 19e–g, Supplementary Fig. 20, b). These data indicate that CaHSF8 but not CaKAN3 acts positively in pepper thermotolerance, consistent with the fact that the majority of HSFs have been implicated in plant thermotolerance[70,71].

By ChIP-seq, a subset of HSP genes, including *CaHSP17.4B*, *CaHSP18.2*, *CaHSP70*, and *CaHSP70-15*, were found among the potential target genes in pepper plants upon extremely high temperature (42 °C) treatment, with the enrichment of CaHSF8 in their promoter regions (Supplementary Fig. 20e). By ChIP–qPCR, these HSP genes were found to be directly targeted by CaHSF8 but not by CaKAN3 (Supplementary Fig. 20f). Consistently, the tested HSP genes were upregulated by CaHSF8 transient overexpression but not by CaKAN3 transient overexpression (Supplementary Fig. 20g). In contrast, *CaKAN3* silencing did not affect the binding of CaHSF8 to the promoters of the four HSP genes (Supplementary Fig. 20h). All these data indicate that HSP and NLR genes are differentially targeted by CaHSF8 in a context-specific manner, and CaHSF8 acts positively in pepper thermotolerance by directly activating at least the tested HSP genes independently of CaKAN3.

### CaHSF8 differentially associates with CaKAN3 in a temperature-dependent manner to coordinate the activation of high-temperature-high-humidity-specific immunity and thermotolerance

As CaHSF8 associates with CaKAN3 to activate pepper immunity upon RSHT but acts alone to activate HSP genes at the later stage of HTHH treatment and extremely high temperature treatment, we speculate that this difference might be attributed to the differential association between CaHSF8 and CaKAN3. To test this possibility, we studied the effect of temperature on the CaHSF8/CaKAN3 interaction by BiFC (Supplementary Fig. 21a) and Nluc/Cluc assays (Supplementary Fig. 21b) and found that the interaction occurred at 1 hpt at 28 °C, 34 °C and 37 °C and at 1 and 3 hpt of 37 °C treatment but not at 6 hpt, while it did not occur at all of the tested time points of 45 °C treatment (Fig. 5a, b). The same result was found in MST (Fig. 5c) and CoIP assays (Fig. 5d, e), indicating that CaHSF8 associated specifically with CaKAN3 at the early stage of HTHH treatment. Consistently, compared to the treatment of RTHH (28 °C, 90% humidity), the activation of the tested 6 NLR genes by CaHSF8 was inhibited upon exposure to 45 °C, 90% humidity or to HTHH (37 °C, 90% humidity) after 3 hpt (Fig. 5f). In contrast, four HSP genes, *CaHSP17.4*, *CaHSP18.2*, *CaHSP70* and *CaHSP70-5*, were upregulated at 45 °C or after long-term treatment at 37 °C (Fig. 5f). In addition, we found that CaHSF8 transient overexpression upregulated the four HSP genes from 1 to 6 hpt in HTHH- or RSHT-challenged

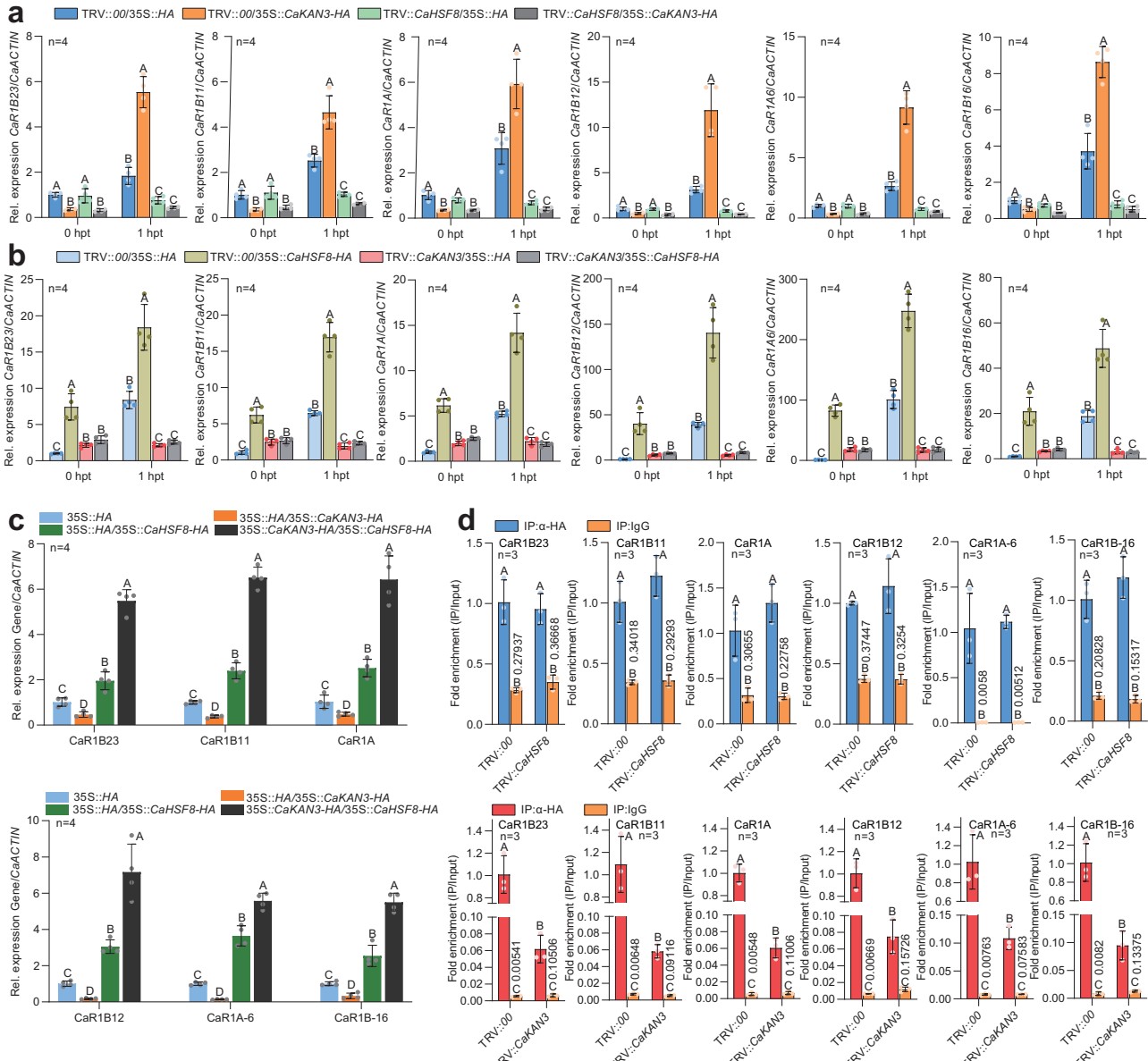

**Fig. 4 | Relationship between *CaKAN3* and *CaHSF8* in the regulation of the six tested NLRs at 1 hpt of HTHH treatment. a, b** The 6 tested NLR genes, *CaR1B23*, *CaR1B11*, *CaR1A*, *CaR1B12*, *CaR1A6* and *CaR1B-16*, were upregulated by CaHSF8 transient overexpression, but this upregulation was blocked by *CaKAN3* silencing in pepper plants at 1 hpt of HTHH, and vice versa. The 6 NLR genes were down-regulated by CaKAN3 transient overexpression at 0 hpt and by *CaHSF8* silencing, and CaKAN3 and CaHSF8 functioned independently at 0 hpt of HTHH. **c** The co-transient overexpression of CaKAN3 and CaHSF8 induced higher expression levels of the 6 tested NLR genes than transient overexpression of CaKAN3 and CaHSF8

individually at 28 °C. **d** By ChIP–qPCR, *CaKAN3* silencing significantly reduced the enrichment of CaHSF8 on the promoters of the tested NLR genes, but *CaHSF8* silencing did not reduce that of CaKAN3 on these promoters. The enrichment of IP:anti-HA was set to 1 after normalization by input. The ratio of IP:anti-HA to IP:IgG is indicated on the error line of IP:IgG. In **a**–**c** *CaActin* was used as an internal control. Data are shown as the means ± standard errors of four replicates. Different uppercase letters above the bars indicate significant differences (*P* < 0.01) by Fisher's protected LSD test. All replicates were from different plants. In **a**–**d**, source data are provided as a Source Data file.

pepper plants (Supplementary Fig. 22a, b). Furthermore, we found that CaHSF8 bound the promoters of the six NLRs with slightly different binding affinities at 28 °C, 90% humidity and 37 °C, 90% humidity but did not bind the promoters of NLR genes at 45 °C, 90% humidity (Supplementary Fig. 23a, b). However, CaHSF8 bound the promoters of the HSP genes at 6 hpt of HTHH (37 °C, 90% humidity) treatment (Supplementary Fig. 23a) or at all of the tested time points upon extremely high temperature and high humidity (45 °C, 90% humidity) (Supplementary Fig. 23b). CaKAN3 bound the promoters of 6 tested NLR genes at 28 °C, 90% humidity, 37 °C, 90% humidity and 45 °C, 90% humidity with slightly different affinities under the three tested conditions, but no significant difference was found between the

treatments and two time points (1 and 6 hpt) (Supplementary Fig. 23 c and d). All these data indicate that high-temperature-specific immunity and thermotolerance are activated by CaHSF8 through its temperature-dependent differential association with CaKAN3.

## Discussion

Despite the importance of plant-specific immunity against RSHT and its distinct nature from that under room temperature[15], the underlying mechanisms remain poorly understood. Herein, our results demonstrate that the immunity of pepper against RSHT was reduced by silencing *CaKAN3* or *CaHSF8*, and their ectopic overexpression promoted *Nicotiana benthamiana* resistance to RSHT. The pepper lines

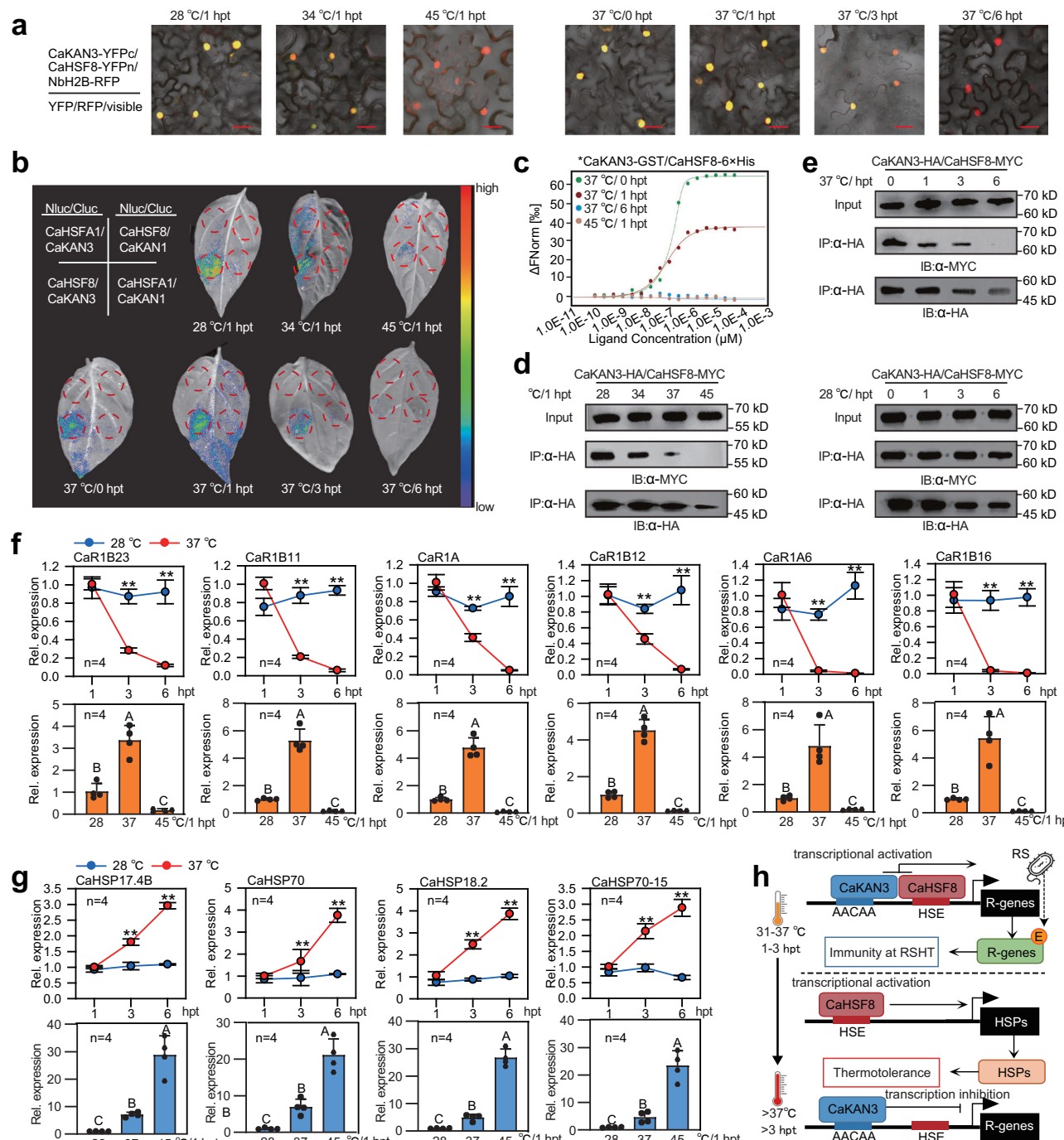

**Fig. 5 | CaHSF8 was differentially associated with CaKAN3 in a temperature-dependent manner to coordinately activate NLR genes and HSPs. a** The data from BiFC showed that the interaction between CaKAN3 and CaHSF8 weakened as the temperature rose from 28 to 45 °C, and this interaction also weakened at 37 °C with time from 0 to 6 hpt. The experiment was carried out once. **b** Analysis of the interaction intensity between CaKAN3 and CaHSF8 under treatment at 28, 34, or 45 °C at 1 hpt and 1, 3, or 6 hpt at 37 °C. CaKAN3 fused with the C-terminus of Luc (LUC^c) was coexpressed with CaHSF8 fused with the N-terminus of Luc (LUC^N) in *N. benthamiana* leaves by GV3101 cells carrying different plasmids (using CaHSFA1-LUC^N and CaKAN4-LUC^c as negative controls). **c** CaKAN3-GST exhibited a reduced binding affinity to CaHSF8 at 37 °C/1 hpt compared to 37 °C/0 hpt, and CaKAN3-GST did not bind CaHSF8 at 37 °C/6 hpt or 45 °C/1 hpt by MST assay. **d**, **e** Analysis of the interaction intensity between CaKAN3 and CaHSF8 in vivo under treatment at 28, 34, 37, or 45 °C at 1 hpt and under treatment at 37 °C at 1, 3, and 6 hpt, as determined by co-IP assay. The same amounts of proteins isolated from pepper

leaves transiently overexpressing CaHSF8-Myc and CaKAN3-HA were used, and the interacting partners of CaHSF8 were immunoprecipitated with an antibody against Myc. The presence of CaKAN3 in the protein complex was assayed by western blotting using an antibody against HA. **f** The transcript levels of the 6 tested NLR genes under transient overexpression of CaHSF8 at 1, 3 and 6 hpt of 28 or 37 °C treatment and under treatment at 28, 37 or 45 °C at 1 hpt. **g** Transcript levels of HSPs under transient overexpression of CaHSF8 under 28 or 37 °C treatment at 1, 3 or 6 hpt and under 28, 37 or 45 °C treatment at 1 hpt. **h** The mechanisms controlling the coordinative and context-specific activation of high-temperature-specific pepper immunity against RSI and thermotolerance mediated by differential CaKAN3-CaHSF8 association. In **f** and **g**, data are shown as the means ± standard errors of four replicates. Different uppercase letters or asterisks above the bars indicate significant differences ($P < 0.01$) by Fisher's protected LSD or t test. Source data are provided as a Source Data file. In **d** and **e**, the experiment was carried out twice with similar results. All replicates were from different plants.

with high levels of RSHT resistance exhibited higher levels of *CaKAN3* or *CaHSF8* transcripts than the lines with lower levels of RSHT resistance. All these data consistently indicate that CaKAN3 and CaHSF8 act positively and synergistically to activate pepper immunity against RSHT, as plant immunity has been generally found to be repressed by high temperature stress[38], and the susceptibility of different pepper inbred lines to RSHT was significantly potentiated by HTHH. It can be concluded that immunity mediated by CaKAN3 and CaHSF8 compensates for some of the immunity repressed by HTHH. CaKAN3 and CaHSF8 were transcriptionally upregulated in different pepper inbred lines, and their ectopic overexpression in *Nicotiana benthamiana* led to enhanced immunity against RSHT, indicating that immunity against RSHT mediated by CaKAN3 and CaHSF8 is conserved in different pepper lines and probably in other solanaceaes, including *Nicotiana benthamiana* plants. These findings are novel since KAN proteins have been implicated mainly in the regulation of plant development[72–74]. The function of CaKAN3 and CaHSF8 in pepper immunity against RSHT is closely linked to GST-encoding genes, including *CaMgst1* and *CaPRP1*. These GSTs were previously found to be modulated by cytokinin signalling and act positively and specifically in pepper immunity against RSHT in our previous study[15]. These results are supported by the results from other plant species showing that other GSTs are crucial for plant disease resistance[75]. Given that CaKAN3 is structurally similar to its orthologs only in the *Capsicum* genus (Supplementary Fig. 1), the role of KAN3 in immunity under HTHH likely exists specifically in the *Capsicum* genus.

Effector-triggered immunity has frequently been found to be repressed by high temperature or high humidity by repressing NLRs[7,37–40]. Although NLR proteins in solanaceous plants, such as Tsw[76] in pepper, Bs4[77], Sl5R-1[78] and I2[79] in tomato, and ZAR1[80] and Roq1[81] in *Nicotiana benthamiana*, have been functionally characterized in ETI, the effect of high temperature or high humidity on NLR-mediated immunity in solanaceous plants is currently unclear. Similar to the results that Sr21 and Sr13 in wheat[82,83], Xa7 in rice[84] and NDR1 in Arabidopsis[85] confer plant immunity at elevated temperatures, our data closely linked CaKAN3 and CaHSF8 with a series of NLRs in pepper immunity against RSHT (Supplementary Fig. 16). This mechanism appears to exist in *Nicotiana benthamiana* since the ectopic overexpression of CaKAN3 or CaHSF8 also enhanced *Nicotiana benthamiana* resistance to RSHT by activating a similar set of NLR genes. The result that a set of NLR genes was upregulated by the CaKAN3-CaHSF8 module is consistent with the result that transcriptional control of plant NLRs is crucial for plant immunity[86,87] and that NLR genes have been found to confer basal immunity or broad-spectrum resistance[88,89]. In addition, CaKAN3 and CaHSF8 act positively in response to RSHT in different pepper lines (Supplementary Fig. 9) and to inoculation of different *Ralstonia solanacearum* strains as well as *Pst DC3000* under HTHH (Supplementary Fig. 10). All these results are reminiscent of synergistic relationships among the signalling sectors in PTI[90] and support the continuum between PTI and ETI[91,92]. The synergistic relationship among the NLR proteins was further supported by the data that they probably form a resistosome-like complex[93] at least with CaR1B-11 interacting with the other five NLR proteins (Supplementary Fig. 16), although the biological consequence of this complex and the underlying molecular details are currently unknown. Thus, we speculate that plants might have evolved specific NLR genes for HTHH-specific immunity during their coevolution with pathogens under HTHH conditions.

Importantly, we established for the first time the association between CaKAN3 and CaHSF8 in activating NLRs and thus pepper immunity against RSHT, even though it has been found that KAN4 interacts with AUXIN RESPONSE FACTOR 3[53] and HSFs associate with a plethora of proteins[94,95]. This interaction enables CaKAN3 and CaHSF8 to synergistically activate the six tested NLR genes, with CaHSF8 being recruited to these target genes by CaKAN3 and with CaKAN3 being transformed from a negative to a positive regulator of the NLR genes

(Figs. 3–5). Similarly, WRKY70 was previously found to be turned from a negative regulator of SARD1 expression to a positive regulator by phosphorylation[96]. One explanation for this is that the role of CaKAN3 as a repressor might be due to modification by some unidentified regulatory proteins, which might be removed or counteracted by CaHSF8 via physical interaction when the plants were challenged by RSHT. It is worth noting that compared with that at 28 °C, the binding of CaHSF8 to NLR promoters was slightly enhanced at 37 °C, which may be attributed to its enhanced recruitment to the tested NLR genes by CaKAN3 that was upregulated at 37 °C. On the other hand, there was only a slight difference in the binding of CaKAN3 to the tested NLR promoters between 28 °C and 37 °C, but this difference was not regular and was not affected by CaHSF8 (Supplementary Fig. 23). In addition, the CaKAN3/CaHSF8 interaction and thus upregulation of the tested NLR genes were found at the early stage (1 to 3 hpt) of HTHH treatment. The upregulated NLRs might activate immunity only in response to invading pathogens by perceiving their corresponding ligands[97], while the upregulated NLRs cannot activate any immunity in the absence of pathogens. However, when HTHH exposure is prolonged to 6 hpt or the plants are exposed to extremely high temperatures, CaKAN3 cannot interact with CaHSF8, and thus, NLR genes cannot be activated. The released CaHSF8 might directly upregulate HSP genes, therefore activating thermotolerance. These results indicate that high-temperature-specific immunity against RSI and thermotolerance are context-specifically activated by CaHSF8 via differential interaction with CaKAN3 (Fig. 5g). It is worth noting that although cytokinin signalling has been implicated in plant immunity and in the trade-off between plant immunity and growth/development[21,98], there is no report on its relationship to NLR and KANADI/HSF so far. We speculate that upon the challenge of combined high temperature and high humidity (31–37 °C, 90% humidity), CaKAN3/CaHSF8 is rapidly activated and interacts with each from 1 to 3 hpt in a pathogen-independent manner, which in turn activates NLRs. These NLRs might further activate high-temperature-high-humidity specific immunity in the presence of pathogen infection upon perception of pathogen ligands by activating *CaMgst3* and *CaPRP1*, probably with the action of cyokinin signalling initiated by pathogen infection under high temperature and high humidity conditions partially through chromatin activation[15]. If there is no pathogen infection after 3 h of high temperature and high humidity treatment or under extremely high temperatures (more than 42 °C), CaHSF8 no longer interacts with CaKAN3, and the released HSF8 alone activates HSP genes by directly binding the promoters and thus activates thermotolerance (Fig. 5h). To elucidate the molecular mechanisms underlying the functional association between cytokinin signalling and NLR, further study is required in the future.

Collectively, to compensate for the immunity impaired by conditions of high temperature and high humidity, CaHSF8 acts positively in high-temperature-high-humidity-specific immunity against *Ralstonia solanacearum* infection by upregulating a subset of NLR genes through interaction with CaKAN3. However, under extremely high temperature conditions or prolonged exposure to conditions of high temperature-high humidity without pathogen infection, CaHSF8 alone activates thermotolerance by upregulating HSP genes.

## Methods
### Plant materials and growth conditions
The seeds of the pepper inbred lines HN42 and *Nicotiana benthamiana* were sown in a soil mix [peat moss: perlite, 2:1 (v/v)] (PINDSTRUP, Denmark) in plastic pots (7 cm × 7 cm × 7.3 cm) and were placed in a growth room under conditions of 28 °C, 60–70 mmol photons m$^{-2}$ s$^{-1}$, a relative humidity of 70%, and a 16-h light/8-h dark cycle.

### Vector construction
The full-length cDNAs of *CaKAN3*, *CaKAN4* and *CaHSFA1*,*CaHSF8* were amplified by PCR using specific primer pairs (Supplementary Table 1)

and then cloned into the entry vector pDONR207 by BP reaction using a Gateway cloning system (Invitrogen, 11789020). To construct the *CaKAN3* or *CaHSF8* destination vectors for the overexpression assay or prokaryotic expression, the full-length *CaKAN3* and *CaHSF8* genes were cloned into pEarleyGate plasmid vectors: pEarlyGate101, pEarlyGate103, pEarlyGate201, pEarlyGate202[99]; pSPYCE, pSPYNE[100]; pPGCL, pPGNL[101], pDEST-17 (Invitrogen, 11803012) or pDEST-15 (Invitrogen, 11802014) by LR reaction.

To construct the vectors for the VIGS assay, the specific gene fragments of *CaKAN3*, *CaHSF8* or NLRs in their 3′UTRs to avoid the possible silencing of their homologous genes were amplified by PCR using specific primer pairs (Supplementary Table 1), and each fragment was cloned into the entry vector pDONR207 by BP reaction individually and then into the destination vector pPYL279 by LR reaction using the Gateway cloning system (Invitrogen, 11791020).

## VIGS assay
To assess the function of *CaKAN3*, *CaHSF8* or NLRs in the pepper response to RSI or HTHH, the VIGS assay carried out in the present study followed our previous description[102]. Transformation of *Agrobacterium tumefaciens* strain GV3101 cells by cold melting[103]. GV3101 cells harboring pTRV1 were mixed with GV3101 cells harboring pTRV2:00, pTRV2:*CaPDS*, pTRV2:*CaKAN3*, pTRV2:*CaHSF8* or pTRV2:*NLRs* at a 1:1 ratio and then incubated at 28 °C at 60 rpm for 3 h. The mixed bacterial solution was injected into the cotyledons of 2-week-old pepper seedlings, which were placed under 16 °C without light for 56 h.

## RTHH, RSRT, HTHH and RSHT Treatments
*Ralstonia solanacearum* strain FJC100301 was isolated from wilted samples of pepper from Fujian Province (China). Exudates of the stem vascular portion from these plants were purified on tetrazolium chloride medium[104]. The cells of *R. solanacearum* strains FJC100301, FJI470 and GMI1000 were cultured at 200 rpm, 28 °C in SPA medium (200 g of potato, 20 g of sucrose, 3 g of beef extract, 5 g of tryptone, and 1 L of water) and distilled sterile 10 mM $MgCl_2$ for 36 h. Soil-grown pepper and *Nicotiana benthamiana* plants with their roots being mechanically damaged (a blade was used to side-cut the root along the stem) were inoculated through root irrigation with a 0.5 mL cell suspension of *R. solanacearum* ($OD_{600} = 1.0$) for each pot (approximately 1.31 mL per L soil). The plants were placed in an illuminated incubator (60–70 μmol photons m$^{-2}$ s$^{-1}$, 16-h light/8-h dark photoperiod) at either 28 °C (for *R. solanacearum* infection at room temperature) or 37 °C (for *R. solanacearum* infection at high temperature). The soil in the pots was kept at its maximum water-holding capacity, while the humidity was kept at least at 80%. Non-inoculated plants were grown under the same conditions with mechanical root damage but without bacterial inoculation, either at 28 °C (room temperature, high humidity) or 37 °C (high temperature, high humidity). The phenotypes of all of the plants were observed at appropriate time points, and the disease indices of 24 plants were dynamically calculated according to the criteria listed in Supplementary Table 2.

## *Pst DC3000* culture and inoculation
The culture and inoculation methods of *Pst DC3000* followed the method of a previous study[105]. *Pst DC3000* was cultured in King's B media containing rifampicin (25 mg ml$^{-1}$) at 28 °C overnight and resuspended in 10 mM magnesium chloride. Whole plants were inoculated with the bacterial suspension at a final concentration of $OD_{600} = 0.1$ ml$^{-1}$ with 0.02% Silwet L-77 by spraying. At 2 and 4 dpt, disease symptoms were assessed by bacterial population counts (colony-forming unit test).

## LC–MS/MS Analysis
To confirm the interaction protein of CaKAN3, *Agrobacterium tumefaciens* strain GV3101 cells containing *CaKAN3-GFP* were infiltrated into

pepper plant leaves, which were harvested at 48 hpi treatment with RSI under high temperature and high humidity or RSI under room temperature and high humidity and ground into powder in liquid nitrogen. The IP procedure was the same as described for CoIP. Samples were run on SDS-polyacrylamide gel, and gel was stained with Coomassie Brilliant Blue. Afterwards, each gel line was cut into several pieces, and these pieces were kept in separate 2 mL microcentrifuge tubes (Axygen, USA). Gel slices were destained in 50% acetonitrile and incubated for 45 min in 10 mM DTT. Cysteinyl residue alkylation was performed for 30 min in the dark in 55 mM chloroacetamide. After several washes with 25 mM ammonium bicarbonate, 50% acetonitrile gel slices were dehydrated in 100% acetonitrile. Gel pieces were rehydrated with 50 mM ammonium bicarbonate and 5% acetonitrile containing 20 ng/μL trypsin (Pierce), and digestion proceeded overnight at 37 °C. Tryptic peptides were sonicated from the gel in 5% formic acid and 50% acetonitrile, and the total extracts were evaporated until dry.

The isolated proteins were analysed on an LTQ-Orbitrap XL mass spectrometer (Thermo Fisher Scientific, USA), as previously described[106]. The peptides were dissolved in 10 μL of a 10% formic acid solution and then analysed using LC–MS/MS with an online sodium spray ion source. The 5 μL peptide samples were loaded into a trap column (Acclaim PepMapC18, 100 μm × 2 cm; Thermo Fisher Scientific) at a flow rate of 10 μL/min and then subsequently separated on a 60-min gradient in an analytical column (Acclaim PepMapC18, 75 μm × 15 cm). The column flow was controlled at 300 nL/min, and the electrospray voltage was 2 kV. The full scan spectra (m/z 350–1550) were obtained at a mass resolution of 60 K, and HCD MS/MS scans were subsequently performed at a resolution of 30 K with dynamic exclusion for 30 s.

The original mass spectrometry collection files were imported into Proteome Discover 2.4 for retrieval (MS1 tolerance: 10 ppm; MS2 tolerance: 0.05 Da; Missed cleavage: 2). The peptide fragments were searched against the database of pepper Zunla-1 (https://solgenomics.net/ftp/genomes/Capsicum_annuum/C.annuum_zunla/), and protein functions were annotated using a BLAST search of the UniProt database.

## Subcellular localization and bimolecular fluorescence complementation (BiFC) assay
To detect the subcellular localization of CaKAN3 or CaHSF8, GV3101 cells containing *35S:CaKAN3-GFP* or *35S:CaHSF8-GFP* were grown in LB medium with 50 μg/mL kanamycin and 25 μg/mL rifampicin at 28 °C. Then, the cells were collected by centrifugation at 28 °C and 5000 rpm for 10 min. The acquired cells were resuspended in infection buffer (10 mM $MgCl_2$, 10 mM 2-morpholinoethanesulfonic acid (MES), and 200 mM acetosyringone, pH 5.6) to $OD_{600} = 0.6$, and appropriate amounts of GV3101 cells were infiltrated into leaves of *Nicotiana benthamiana* plants. Images were taken at 48 hpi by a laser scanning confocal microscope (TCS SP8; Leica Microsystems, Wetzlar, Germany). To confirm the protein interaction between CaKAN3 and CaHSF8, GV3101 cells containing *pSPYCE-CaKAN3* (using *CaKAN4* as a control) constructs mixed with *pSPYNE-CaHSF8* (using CaHSFA1 as a control) constructs at a 1:1 ratio were infiltrated into *Nicotiana benthamiana* leaves. Images were taken at 48 hpi by a laser scanning confocal microscope (TCS SP8; Leica Microsystems, Wetzlar, Germany), with the excitation wavelength and emission filter 488 nm (GFP), 510 nm (YFP), 587 nm (RFP) and bandpass 500 to 550 nm (GFP and YFP), 585 to 635 nm (RFP), respectively, and the objective was 100×, the scanning frequency was 144 Hz for cell observation and 20 Hz for fluorescence photography, Image size: 150 × 150 μm, Format: 1024 × 1024.

## Prokaryotic expression
To obtain sufficient amounts of soluble CaKAN3 or CaHSF8 protein, pDEST-15 or pDEST-17 plasmids harboring *CaKAN3-GST* or *CaHSF8-6×His* were transformed into the *Escherichia coli* (*E. coli*) strain BL21 (DE3). BL21 (DE3) competent cells and plasmids were exposed to an ice bath for 30 min and heated at 42 °C for 90 s. After 5 min in an ice bath,

LB medium without antibiotics was added and incubated at 37 °C for 1 h. Then, the bacterial solution was coated on solid LB medium containing ampicillin and cultured overnight in the dark at 37 °C. Expression of the fusion protein was induced with IPTG (isopropyl β-D-1-thiogalactopyranoside, 1 mM) at 20 °C for 12 h, and then bacterial cells were collected by centrifugation (4 °C, 12000 rpm), dissolved in precooled 0.1 M PBS, and broken by KLB-UH600 (KEWLAB, P: 200 W, open time: 2 s, stop time: 2 s, all time: 30 min). An SDS–PAGE assay was performed to confirm whether the soluble fusion protein was present in the supernatant of the *E. coli* cell lysate. The electrophoretic SDS–PAGE gel was stained with Coomassie brilliant blue R250 (Sigma-Aldrich, 1.12553) and then decoloured using decolourizing solution (10% acetic acid and 5% ethanol) to remove the background to determine whether the protein was successfully expressed by observing the bands on the gel.

### Coimmunoprecipitation (Co-IP) and western blot analysis

To confirm the interaction between CaKAN3 and CaHSF8, GV3101 cells containing *CaHSF8-GFP* and *CaKAN3-Myc* (*CaHSFA1-GFP* and *CaKAN3-Myc*, *CaHSF8-GFP* and *CaKAN4-Myc* or *CaHSFA1-GFP* and *CaKAN4-Myc*) were mixed at a 1:1 ratio, and the respective mixture was infiltrated into *Nicotiana benthamiana* plant leaves, which were harvested at 48 hpi and ground into powder in liquid nitrogen. Total protein was extracted by 0.1 M PBS solution with a protein inhibitor cocktail (Roche, Basel Swiss) and incubated on ice for 1 h. The protein solution was slowly mixed with GFP-trap beads (Abcam, Shanghai China) and spun down after incubation for 1 h at 4 °C and then eluted with elution buffer after washing 5 times for further use.

To detect the target proteins, a western blot assay was performed[102]. The proteins were boiled with protein loading buffer at 95 °C and then separated by SDS–PAGE. The proteins in the gel were transferred to PVDF membranes (Thermo Fisher Scientific, Waltham, MA, USA) by a semidry transfer system (Bio-Rad, USA) at 200 mA for 30 min. The PVDF membranes were immersed in blocking buffer for 1 h at room temperature and then incubated in a primary anti-Flag antibody (Abmart, Shanghai China) or an anti-GFP antibody (Abcam, Shanghai China) diluted at a 1:5,000 ratio. After washing 3 times with TBST buffer, the PVDF membranes were incubated in a secondary antibody (goat anti-rabbit immunoglobulin G (Sigma–Aldrich) with horseradish peroxidase conjugate) diluted at a 1:50,000 ratio.

### Microscale thermophoresis (MST) analysis

The MST assay was used to confirm the protein–protein interaction as described in previous studies[107]. To confirm the protein interaction between CaHSF8-6×his and CaKAN3-GST, the two fused proteins were prokaryotically expressed, and CaKAN3-GST was marked by fluorescence (Mo-L011, NanoTemper Technologies, Germany). CaHSF8-6×his was diluted to concentrations ranging from 1.0E-10 mM to 1.0E-3 mM and mixed with 20 mM GST or CaKAN3-GST protein solution. The mixtures were incubated with an interaction buffer (100 mM NaCl, 1 mM EDTA, 20 mM sodium phosphate, pH 8.0). The samples were then loaded into Monolith NT.115 Capillaries (MO-K002, NanoTemper Technologies, Germany) using 50% IR laser power and an LED excitation source, where $\lambda = 470$ nm at ambient temperature. The Kd values for the protein interactions were calculated using NanoTemper Analysis 1.2.20 software[108].

### Genetic Transformation of *Nicotiana benthamiana*

Genetic transformation of *Nicotiana benthamiana* followed the method of Regner et al.[109] and Bardonn et al.[110]. Leaf discs of *Nicotiana benthamiana* were transformed with GV3101 cells bearing various vectors (*35Spro:CaKAN3-GFP* or *35Spro:CaHSF8-GFP*). Independent $T_0$ transgenic *Nicotiana benthamiana* plants were selected on 10% PPT (glufosinate, Sigma, 45520) and later confirmed by PCR with specific primers (Supplementary Table 1). $T_0$ plants were self-pollinated, and seeds of each plant were separately harvested. The $T_1$ plants were selected by germination of seeds harvested from $T_0$ plants on MS medium (PhytoTech, M519) supplemented with PPT (1:250), which was further confirmed by PCR using specific primers (Supplementary Table 1). Similarly, seeds of the $T_2$ and $T_3$ lines were acquired, and the homozygous $T_3$ lines were used for functional assays of the tested genes.

### Chromatin immunoprecipitation (ChIP) and ChIP-seq

The ChIP assay was performed according to the protocol of our previous study[34]. The leaves of pepper plants (or VIGS pepper plants) transiently expressing the tested gene by agroinfiltration were crosslinked with 1% formaldehyde for 15 min, and then the cross-linking reaction was terminated with 5 M glycine. The chromatin was isolated following a previously reported method[34], and the acquired chromatin was sheared into fragments of 300–500 bp by sonication (Covaris M220, Sonolab 7.2, ChIP_5%df_6min). The acquired DNA fragments were immunoprecipitated with 5 µg anti-GFP antibody (Abcam, Cambridge, UK) or anti-HA antibody (Abcam, Cambridge, UK, Ab9110). A 5 M NaCl solution and protease K (Invitrogen, RP-87705) were added to the immunoprecipitated sample and incubated overnight at 37 °C. Then, DNA in the sample was purified using precooled DNA extract (phenol:chloroform:isoamyl alcohol = 25:24:1). The purified DNA was precipitated overnight with ethanol (1:3) and dissolved in an appropriate volume of $ddH_2O$. The DNA acquired by immunoprecipitation with different antibodies was purified and used as a template for ChIP–PCR or ChIP–qPCR using specific primer pairs for ChIP–qPCR (Supplementary Table 1).

For ChIP-seq, we infiltrated 30 fully expanded leaves of pepper plants at the 6-leaf stage with GV3101 cells harbouring the binary vector *35 S:CaKAN3-GFP* or *35 S:CaHSF8-GFP*. At 48 hpi, the infiltrated leaves were harvested and cross-linked with 1% formaldehyde, and the chromatin was isolated and subjected to ChIP following the above-mentioned method. The decrosslinked and purified DNA sample was subjected to linear DNA amplification (LinDA) to generate sufficient material to construct ChIP sequencing libraries using an NEBNext ChIP-seq Library Pre Reagent Set for Illumina (New England Biolabs, Ipswich, MA). DNA sequencing was performed on an Illumina HiSeq2500 platform (Novogene, Beijing, China), which resulted in approximately 10 million 100 bp single-end reads per sample. We removed the low-quality reads by fastp software[111], reads with over 15% ambiguous bases, reads contaminated with 5′ barcodes, and reads without 3′ linker sequences or inserts. We also trimmed the 3′ linker sequences and discarded reads shorter than 18 nt after data cleaning. The remaining reads were aligned to the pepper reference genome by BWA (Burrows Wheeler Aligner)[112]. DNA fragment sizes were predicted using MACS2 software, which were then used for subsequent peak analysis. We also used MACS2 software (with threshold $q$-value = 0.05) to detect signal peaks, analyze the number, width, and distribution of peaks, and determine the corresponding genes identified by the peaks[113].

### EMSA analysis

To obtain a large number of pure target proteins and confirm the binding of CaKAN3 or CaHSF8 to the cis-elements within the probe, prokaryotic expression and EMSA were performed following the methods used in our previous study[34]. The wild-type or mutated probe was synthesized by PCR using a single-strand primer and another single-strand primer labeled with Cy5-labeled oligonucleotides. The recombinant proteins of CaKAN3-GST or CaHSF8-GST were incubated with wild-type or mutated probe, which was labeled with Cy5 fluorochrome, and 5× binding buffer (1 M Tris-HCl, pH 7.5, 5 M NaCl, 1 M KCl, 1 M $MgCl_2$, 0.5 M EDTA, pH 8.0, 10 mg/mL bovine serum albumin). The total system was 20 µL. The samples were incubated away from light at room temperature for 1 hour, and then nondenaturing PAGE gel electrophoresis was performed in an ice bath and scanned by Odyssey CLX (LI-COR).

## Total RNA extraction and RT–qPCR assay

One individual plant was harvested for RNA extraction per treatment. The harvested leaves were frozen immediately in liquid nitrogen and transferred to a −80 °C freezer. Frozen leaf tissues were disrupted in 2 mL RNAse-free microcentrifuge tubes (Axygen, USA) using three stainless steel beads and the Tissue Lyser II (Qiagen, Dusseldorf, Germany). TRIzol (Invitrogen, Carlsbad, USA) was used to extract the total RNA from the disrupted tissues (0.5 ml was added to a tube), chloroform was further added to the acquired sample for RNA extraction, isopropyl alcohol was added for RNA precipitation, and ddH2O was added to dissolve the RNA after cleaning the precipitate with 75% ethanol. Both RNA concentration and quality (A260/A280: 1.8-2.2; A260/A230: 1.9-2.1) were confirmed by a NanoDrop 2000 (Thermo Scientific, Massachusetts, USA). The RNA integrity was confirmed by agarose electrophoresis. RNA (500 ng from each sample), 250 ng of oligo dT(15) primer, and 200 unit reverse transcriptase were used for a reverse transcription reaction using One Step PrimeScript™ cDNA Synthesis Kit (TaKaRa, Shigo, Japan), which includes a genomic DNA digestion step, with the following programme: 42 °C, 60 min; 85 °C, 5 s; 4 °C, forever. The synthesized cDNA products were diluted ten-fold for further qPCR analysis. To detect the relative transcript levels of the target genes, a Bio-Rad Real-Time PCR system (Bio-Rad Laboratories, USA) and SYBR Premix Ex Taq (Perfect Real Time; TaKaRa) were used with the specific primer pairs listed in Supplementary Table 1. *CaActin* acted as an internal reference gene to normalize the transcript expression levels, which have been confirmed by a published paper[114]. The Livak method was used to analyse the data[115].

## Split-luciferase complementation assay

The LUC assay was performed as described in previous studies[116]. GV3101 cells containing *35 S:CaKAN3-LUC^c* or *35 S:CaKAN4-LUC^c* constructs mixed with *35 S:CaHSF8-LUC^N* or *35 S:CaHSFA1-LUC^N* constructs at a 1:1 ratio were infiltrated into leaves of *Nicotiana benthamiana* plants, which were incubated in a growth room for 48 h. Luciferin (1 mM) was sprayed onto the leaves for CCD imaging (Charge coupled Device, Nightshade LB985).

## DAB (diaminobenzidine) staining and chlorophyll fluorescence spectrophotometry

DAB staining was used to assess $H_2O_2$ accumulation in pepper leaves, as described previously[34]. For the thermotolerance assay, overexpressing *Nicotiana benthamiana* leaves or VIGS leaves were placed in an illumination incubator at 42 °C and 90% relative humidity for 48 h, one mature leaf from the same part of each plant was collected as a biological repeat, and collected leaves were immersed in 1 mg/ml DAB (Sigma, D12384) solution for 24 h in the dark. Then, 75% ethanol was used to remove the background. To measure thermotolerance displayed by $F_v/F_m$ and $\triangle F/F_m'$ values, the leaves of pepper or *Nicotiana benthamiana* were adapted to darkness for 15 min and were placed into the instrument for measurement according to the method of Schreiber[117] using a MINI Imaging PAM instrument (Heinz Walz GmbH, Effeltrich, Germany) in pepper and *Nicotiana benthamiana* leaves.

## GUS activity assay

We extracted total proteins with GUS protein extraction buffer (0.1 M phosphate-buffered saline (PBS) pH 7.0, 10% SDS, 0.5 M EDTA pH 8.0, 0.2% Triton X-100, 0.2% β-mercaptoethanol)) and measured GUS activity using 4-methylumbelliferyl β-D-glucuronic acid (4-MUG) as a substrate according to Jefferson[118], and the total protein concentration was determined by the Bradford method. To measure the concentration of the GUS product 4-methylumbelliferyl (4-MU) resulting from the conversion of 4-MUG, we added 5 µL of total protein extract to 195 µL of GUS assay buffer (0.44 mg/mL 4-MUG in GUS extraction buffer), incubated the reactions at 37 °C and measured 4-MU fluorescence (excitation: 365 nm; emission: 455 nm) after 10, 20, 30, and 60 min in a microplate reader (Synergy H1, Biotek). GUS activity was expressed in units of pmol 4-MU produced per milligram of protein per minute.

## ROS measurements

Leaf discs of $0.25\ cm^2$ were excised from 5-week-old pepper plants, followed by an overnight incubation in a 96-well plate with 200 µl of $H_2O$. $H_2O$ was replaced by 100 µl of reaction solution (20 µM luminol, $1\ µg\ ml^{-1}$ horseradish peroxidase) supplemented with 500 nM flg22 (P6622; PhytoTech). Reactive oxygen species measurements were conducted immediately using a luminometer (GM2000; Promega) with a 1-min interval reading time over a period of 40 min.

## Statistics and reproducibility

Statistical analyses of the bioassays were performed with DPS software package. Statistical tests used were all two-sided. For the comparison between multiple groups of samples, data represented the means ± SD obtained from three, four or eight replicates; different uppercase letters above the bars indicated significant differences among means ($P < 0.01$), as calculated with Fisher's protected least-significant-difference (LSD) test. The t-test was used to compare the two groups of samples, data represented the means ± SD obtained from three, four or eight replicates; asterisk above the bars indicated significant differences among means ($P < 0.01$).

## Reporting summary

Further information on research design is available in the Nature Portfolio Reporting Summary linked to this article.

## Data availability

The data supporting the findings of this work are available within the paper and its Supplementary Information files. A reporting summary for this article is available as a Supplementary Information file, the ChIP-seq data reported in this paper have been deposited in the Genome Sequence Archive (GSA) in National Genomics Data Center (NGDC) under accession number CRA011827. Source data are provided with this paper.

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

## Acknowledgements

This work was supported by grants from the National Natural Science Foundation of China (31572136, 31372061) and the Development Fund Project of Fujian Agriculture and Forestry University (CXZX2016158, CXZX2017548). We thank Mark D. Curtis for kindly providing the Gateway destination vectors and Dr S. P. Dinesh-Kumar of the University of California for the pTRV1 and pTRV2 vectors. We thank Huasong Zou and Wei Wu of the Fujian Agriculture and Forestry University for kindly providing the *Pst DC3000*.

## Author contributions

S.L.H. and S.Y. conceived the research and designed the experiments. S.Y., W.W.C., R.J.W., Y.H., Q.L.L., H.W., X.Y.H., Y.P.Z., Q.W., X.G.C., M.Y.W. J.G.L., Q.L., and X.Z. performed the experiments. S.Y., S.L.M., and D.Y.G. analyzed the data. S.L.H. wrote the manuscript. All authors of this paper read and approved the final manuscript.

## Competing interests

The authors declare no competing interests.
