## [Peer Review File · Nature Communications]

REVIEWER COMMENTS

Reviewer #1 (Remarks to the Author):

The manuscript (Yang et al.) describes the roles of CaKAN3 and CaHSF8 in high temperature-dependent expression of some R gene-mediated immunity against *Ralstonia solanacearum* in pepper. CaKAN3 gene was initially identified as an up-regulated gene from pepper roots challenged with *Ralstonia solanacearum* at high temperatures and VIGS-mediated knock-down of CaKAN3 in HN42 inbred pepper plants resulted in an effect pepper defense against *Ralstonia solanacearum* at high temperatures (37°C), but not at ambient temperature (28°C). As an interacting partner, CaHSF8 protein was identified through IP-MS analysis and further analysis revealed that loss of function of CaHSF8 also showed compromised defense against *Ralstonia solanacearum* at high temperatures (37°C), but not at ambient temperature (28°C). Through ChIP-sequencing results using transiently expressed CaKAN3 and CaHSF8 in pepper plants, six NLR genes were identified as among the target genes and the loss of function of each of the six NLR genes compromises immunity against *Ralstonia solanacearum* at high temperatures. Finally, prolonged high temperatures (more than 6h at 37°C) or extreme temperatures (e.g., 45°C) compromise the interaction between CaKAN3 and CaHSF8 in vitro and in vivo conditions and expression of six target NLR genes.

The manuscript was easy to follow. Most of the presented datasets and conclusions look reasonable. We do have several significant issues for the authors.

Major points

1. A main criticism of this work is that the writing is misleading. Based on data in Fig. 1, high temperature (HT) GREATLY suppresses pepper resistance against *R. solanacearum* overall. It is only in this broad context HT-mediated immune suppression that CaKAN3, CaHSF8 and the six NLR genes SLIGHTLY increase pepper resistance. This needs to be clearly stated in the abstract (i.e., HT greatly suppresses pepper resistance against *R. solanacearum* overall) and/or discussion (i.e., the same point above with reference to the moderate increase of resistance conferred by CaKAN3/CaHSF8 and recent publications on the overall negative effect of high temperature on plant disease resistance).
2. It seems odd that silencing each of six HT-regulated NLR genes is sufficient to completely compromise HT-mediated resistance. The authors did not discuss this rather strange result based on our current understanding of NLR gene function and signal transduction, unless the products of these six NLR genes form a single complex. Otherwise, one would expect partial or no effect of some NLR genes.
3. The authors also did not discuss if the CaKAN3 or CaHSF8-mediated thermo-tolerant immunity is applicable to other inbred lines of pepper. Is the CaKAN3 or CaHSF8-mediated immunity at high temperatures a conserved mechanism in diverse inbred lines of pepper plants? Or is it a specific mechanism in the HN42 inbred line? For example, the HN42 inbred line is relatively more tolerant to *Ralstonia* infection at high temperatures than other inbred lines used in the previous publication (Yang et al. 2022 *Plant Cell Environ*). Do other tolerant inbred lines against *Ralstonia* infection at high temperatures have CaKAN3 or CaHSF8-mediated thermo-tolerant immunity? In addition, could over-expression of CaKAN3 or CaHSF8 confer temperature-tolerant immunity to temperature-susceptible inbred lines?
4. In Supplementary Data Fig.4b and 5j, 35S::CaKAN3-GFP and 35S::CaHSF8-GFP transgenic *Nicotiana benthamiana* plants are more tolerant to *Ralstonia* infection at high temperatures. Does *Nicotiana benthamiana* plant have CaKAN3 and CaHSF8 homologs in its

genome? Are NLR genes regulated by CaKAN3 protein conserved in *Nicotiana benthamiana*?

5. Is thermo-tolerant immunity through CaKAN3 and CaHSF8 affect basal immunity. In Fig. 1e and Supplementary Data Fig. 5g, flg22-mediated ROS was compromised in knock-down plants of CaKAN3 and CaHSF8 genes. Can you test the phenotypes of CaKAN3 or CaHSF8 knock-downed plants against other virulent *Ralstonia solanacearum* stains?

6. There are two different size bars on imaging data in Fig.2c, Supplementary Data Fig.3, and Supplementary Data Fig.5b. Also, the magnification of each image varies. Please use equal and consistent imaging conditions for a better comparison.

7. Please include control experiment results showing the expression of proteins in BiFC and split Luciferase experiments (Fig2c, Fig 5a).

Minor points

1. The image used in Supplemental Data Figure 5 is too small.

2. Typos need to be fixed (line 155)

3. Please replot ChIP-qPCR results in Fig3e and Fig4d to show the relative abundance of IP:HA samples against IP:IgG controls.

4. The font size is too small in several figures.

5. Please provide uncropped gel/immunoblotting images.

6. Please spell out abbreviations (line 112, 134, 137, 139, 172).

Reviewer #2 (Remarks to the Author):

Dear Editor/Authors,

Yang and co-authors have conducted a detailed and comprehensive investigation into the roles of pepper KAN3 and HSF8 (and their interaction) in temperature-specific disease resistance and thermotolerance. The authors showed that heat-induced KAN3 is a positive regulator of pepper immunity specifically at high temperature (37C) but not at the control temperature (28C). Additionally, they demonstrated that KAN3 interacts with HSF8 in vitro and in vivo, while also dissecting the KAN3 and HSF8 protein domains mediating this interaction. In terms of transcriptional regulation, KAN3 and HSF8 ChIP-Seq and EMSA showed enriched recruitment to immunity-associated intracellular immune receptor (NLR) genes, which exhibit enhanced expression at 37C. Mechanistically, loss of KAN3 reduces HSF8 binding to target NLR promoters and vice-versa. The authors then attempt to reveal a novel context-specific mechanism, wherein higher heat shock temperatures (42C) reduce NLR gene expression but induce HSP gene expression via thermosensitive recruitment of HSF8.

This study is quite novel and exciting as it presents detailed characterization of agriculturally important solanaceous plants using both VIGS and overexpression analyses. Comprehensive (and complementary) assays are also performed at the molecular,

biochemical and physiological levels. Overall, the research presented in this manuscript should be significant to a broad readership, since it is at the nexus of gene regulation, plant immunity and plant stress biology.

To improve the manuscript, some issues need to be addressed and/or clarified:

1) The central conclusion of the paper is the differential interaction between KAN3 and HSF8 with each other and with target defence promoters under different temperatures. This is shown in the authors' model in Figure 5g. However, direct evidence on the main thesis of the paper is a bit lacking in my opinion. Can the authors directly demonstrate that KAN3-HSF8 protein interaction is differentially regulated by temperature using BiFC at 28C vs. 37C vs. 42C/45C.

2) Although the authors show HSF8 ChIP analyses of target NLR and HSP genes at different temperatures (Supplementary Figure 14), the corresponding KAN3 ChIP under different temperatures (28C, 37C and 42C) seems to be missing. Supplementary Fig 7 only shows KAN3 promoter binding dataset at one temperature. To fully support the authors conclusions and claims, I believe that this matter needs to be clarified.

3) The authors conclude that temperature-regulated HSF8 recruitment leads to thermosensitive NLR gene expression. However, there is no difference in HSF8 recruitment to 4 NLR genes (CaR1B23, CaR1B11, CaR1A, CaR1B12) between 28C and 37C (Supplementary Figure 14b). Differential HSF8 recruitment at 28C vs. 37C is only observed in two NLR genes CaR1A6 and CaR1B16. How do the authors reconcile these with their data showing upregulated expression of all 6 NLR genes at 37C (Figure 4a-b)? I believe that this needs to be properly discussed in the paper.

4) The Discussion section needs to synthesize the authors' comprehensive findings and then properly relate it to the broader and relevant literature. How do KAN3 and HSF8 directly connect to the cytokinin-mediated immunity required at high temperature that the authors previously discovered (Yang et al., 2021 Plant, Cell & Environment)? Do KAN3 and HSF8 directly regulate Mgst3 and PRP1 by binding those gene promoters as well?

5) In relation to #4, why did KAN3 not activate transcription in the GAL4 Y2H? Is this somehow related to its putative role as a transcriptional repressor? If KAN3 belongs to a family of transcriptional repressors, how does this link with the current study showing transcriptional activation of NLR genes by KAN3? Does KAN3 switch between repressor and activator activities depending on temperature and/or protein interactors? Perhaps there could be a discussion of other transcription factors that exhibit this context-specific functional switch.

6) Although the methodology is sound and comprehensive, I believe additional experimental details are needed to ensure future reproducibility by other researchers.

- Plant materials: Please indicate the supplier for the soil mix and the dimensions of the pots.
- Vector Construction: Please provide sources/citations for the plasmids. If applicable, indicate the AddGene number.
- VIGS assay: How was GV3101 transformed with the VIGS or empty vector? Please state or add a reference.
- Bacterial treatments: Please detail how *R. solanacearum* strain Fjc100301 was cultured and prepared for infection assays. Also reference the source for the bacterial strain.
- LC-MS/MS analysis: How was the IP portion of the IP-MS experiment performed? How was phosphopeptide enrichment performed? Was the "match between runs" feature used during the MaxQuant search? How was the data normalized for the proteomics analysis?

- **Subcellular localization and bimolecular fluorescence complementation (BiFC) assay:** What were the specific components of the agroinfiltration buffer? How were GV3101 cells prepared (media and growth conditions)? What settings were used for the laser scanning confocal microscope in subcellular localization and BiFC experiments? For example, what were the excitation and emission settings? What time of scanning method was used and how long? What objective lens was used? What was the voxel size and area size for scanning?
- **Prokaryotic Expression:** How was *E. coli* transformation conducted? How was SDS-PAGE of bacterial proteins done? What supplies/conditions were used? Was it the same as in the next subsection of your Methods? Please provide details.
- **Coimmunoprecipitation (Co-IP) and western blot analysis:** What bacterial cells (Line 415) were used? Please be specific.
- **Genetic Transformation of *Nicotiana benthamiana*:** Indicate the supplier of glufosinate used. Specify components, concentration and suppliers of the MS media.
- **Chromatin immunoprecipitation (ChIP) and ChIP-seq:** How was the fixation quenched (e.g. glycine? What concentration?)? Please include the brand and settings of the sonicator. How much antibody was used for ChIP? How was the ChIP DNA purified in detail? What software or R package was used to remove the low-quality reads and to trim the 3' linker sequences?
- **EMSA analysis:** Please provide details in the Supplementary Information for easier reproducibility by other researchers.
- **RT-qPCR assay:** How was total RNA extracted? What type of QC was performed (A260/A280, A260,230, RIN scores, etc.)? What cutoff was used for subsequent gene expression analyses? How was cDNA synthesis performed? Please cite other studies that used CaActin as an internal control gene.
- **DAB staining and chlorophyll fluorescence spectrophotometry;** Provide details on how warm water treatment was performed on the plants. How old were these plants? Were they directly submerged individually or in plates?
- **ROS measurements:** Indicate the supplier of flg22.

7) Reading through the text, the use of abbreviations can be toned down. I was a bit overwhelmed by the ubiquitous use of acronyms (RTHH, HTHH, RSRT, RSHT, ETHH, etc.). Would it be possible to rewrite the manuscript to minimize acronym usage and just state the actual temperature used? This would improve the article's readability.

Figures and Tables:

- **All figure captions (including supplementary figures):** Description of the experimental design and statistical analyses should be further clarified. Please define what the sample n (biological replicates) means. Are they individual leaves from different plants? Are they pooled leaves from different plants? Are they different leaves but from the same plant?
- **All figure captions (including supplementary figures):** For every panel, please indicate the exact number of experimental repeats conducted that led to reproducible results shown by the representative figure.
- **Figure 2e:** the anti-His bands for the EV 6X-His setup is missing in the Input immunoblot. Please make sure to show them and avoid overcropping of the blot.
- **Figure 2f:** The KAN4 IP shows pulldown of HSF8 but not the KAN3 IP. Was this switched because it is not consistent with the authors' conclusions indicating KAN3-HSF8 interaction?
- **Figure 3f:** The blots seem eerily similar even though they are separate experiments with different probes. To be safe, please recheck this figure for potential/inadvertent image duplications.
- **Supplementary Figure 2:** The RTHH acronym is not defined in the caption.

- **Supplementary Figure 4b:** Please add the time course disease index data (similar to Supplementary Figure 5e)
- **Supplementary Figure 4c:** Please indicate which is RSRT and RSHT.
- **Supplementary Figure 5k:** Please indicate which is RSRT and RSHT.
- **Supplementary Figure 6c:** Because the authors are comparing between two temperatures, the statistical analyses should include the comparison between RTHH and HTHH. Also, the caption should state "yeast one-hybrid" not "yeast two-hybrid."
- **Supplementary Figure 11:** Quantify the survival rates and perform statistical analyses in Panel a. Also, the acronym HTS is not defined.
- **Supplementary Figure 14b:** Statistical analyses of the graphs are missing.
- **Supplementary Table 1:** Indicate the sources for all primers or probes (i.e. previous publication or this study?)
- **Supplementary Table 2:** Reference the research that first used this disease index. Or was this implemented for the first time in this study?

Text:

- **Line 37;** Make sure to cite your opening sentence, perhaps using articles that have examined at the impact of temperature and humidity on plant diseases. Generally, it would be good to expand the literature context in this paragraph. That is, I would recommend synthesizing what is currently known on how environmental factors influence plant immunity and plant-pathogen interactions.
- **Line 41:** This sentence should have a citation.
- **Lines 46-61:** Because your study looks at pepper disease resistance, I would recommend also citing literature on plant immune receptors and signaling specifically in solanaceous plants.
- **Line 60-61:** This phrase should be revised in the context of pepper plants: "how TFs function in plant immunity against pathogens under HTHH is unclear." There have been some studies on transcription factors involved in temperature-regulated immunity in other plants (like Arabidopsis), e.g. CAMTA3, PIF4, CBP60g, SARD1.
- **Lines 93-99:** This is one long sentence that could negatively affect readability. I would suggest breaking this up into a few shorter sentences.
- **Line 103:** I would suggest stating *Nicotiana benthamiana* instead of NB since the paper already has lots of acronyms. This would enhance readability.
- **Lines 314-316:** Please also discuss studies showing temperature regulation of solanaceous NLRs and ETI (e.g. in tomato, tobacco, etc.) since your study deals with Solanaceae.
- **Line 317:** It would be good to relate this to studies showing more effective NLR-mediated ETI at high temperatures, e.g. Sr21, Sr13 in wheat and Xa7 in rice.
- **Line 486:** Please define the acronym CCD.

REVIEWER COMMENTS

Reviewer #1 (Remarks to the Author):

The manuscript (Yang et al.) describes the roles of CaKAN3 and CaHSF8 in high temperature-dependent expression of some R gene-mediated immunity against *Ralstonia solanacearum* in pepper. CaKAN3 gene was initially identified as an up-regulated gene from pepper roots challenged with *Ralstonia solanacearum* at high temperatures and VIGS-mediated knock-down of CaKAN3 in HN42 inbred pepper plants resulted in an effect pepper defense against *Ralstonia solanacearum* at high temperatures (37°C), but not at ambient temperature (28°C). As an interacting partner, CaHSF8 protein was identified through IP-ms analysis and further analysis revealed that loss of function of CaHSF8 also showed compromised defense against *Ralstonia solanacearum* at high temperatures (37°C), but not at ambient temperature (28°C). Through CHIP-sequencing results using transiently expressed CaKAN3 and CaHSF8 in pepper plants, six NLR genes were identified as among the target genes and the loss of function of each of the six NLR genes compromises immunity against *Ralstonia solanacearum* at high temperatures. Finally, prolonged high temperatures (more than 6h at 37°C) or extreme temperatures (e.g., 45°C) compromise the interaction between CaKAN3 and CaHSF8 in vitro and in vivo conditions and expression of six target NLR genes.

The manuscript was easy to follow. Most of the presented datasets and conclusions look reasonable. We do have several significant issues for the authors.

Major points

1. A main criticism of this work is that the writing is misleading. Based on data in Fig. 1, high temperature (HT) GREATLY suppresses pepper resistance against *R. solanacearum* overall. It is only in this broad context HT-mediated immune suppression that CaKAN3, CaHSF8 and the six NLR genes SLIGHTLY increase pepper resistance. This needs to be clearly stated in the abstract (i.e., HT greatly suppresses pepper resistance against *R. solanacearum* overall) and/or discussion (i.e., the same point above with reference to the moderate increase of resistance conferred by CaKAN3/CaHSF8 and recent publications on the overall negative effect of high temperature on plant disease resistance).

Response: Thank you very much for this good suggestion, we have rewritten that the high temperature specific immunity mediated by CaKAN3/CaHSF8 compensates partially immunity repressed by high temperature stress following your suggestion in the corresponding parts in the abstract and discussion section, please see line 19 to line 21 in the abstract section and line 404 to line 407 in the discussion section in the revised MS.

2. It seems odd that silencing each of six HT-regulated NLR genes is sufficient to completely compromise HT-mediated resistance. The authors did not discuss this rather strange result based on our current understanding of NLR gene function and signal transduction, unless the products of these six NLR genes form a single complex.

Otherwise, one would expect partial or no effect of some NLR genes.

Response: Thank you very much for this good question, we originally found that the silencing of each of the tested 6 NLR genes produced clearly phenotypic effect on the disease resistance of pepper plants, implying that all of these NLRs are required for their function in pepper immunity in a synergistic manner, and we have performed extra experiments to study the possible interaction in vivo by BiFC and in vitro by MST (Fig S16 f and g), the results showed that R1B-11 interacted with the other five NLR proteins in vivo and in vitro, indicating that these NLRs might form a resistosome like complex by interacting with each other, thus the silencing of each NLR might lead to the functional loss of this resistosome, but the biological consequence and the molecular details remain to be elucidated in the future, please see line 311 to line 320 in the results section and line 436 to line 439 in the Discussion section in the revised MS.

3. The authors also did not discuss if the CaKAN3 or CaHSF8-mediated thermo-tolerant immunity is applicable to other inbred lines of pepper. Is the CaKAN3 or CaHSF8-mediated immunity at high temperatures a conserved mechanism in diverse inbred lines of pepper plants? Or is it a specific mechanism in the HN42 inbred line? For example, the HN42 inbred line is relatively more tolerant to *Ralstonia* infection at high temperatures than other inbred lines used in the previous publication (Yang et al. 2022 *Plant Cell Environ*). Do other tolerant inbred lines against *Ralstonia* infection at high temperatures have CaKAN3 or CaHSF8-mediated thermo-tolerant immunity? In addition, could over-expression of CaKAN3 or CaHSF8 confer temperature-tolerant immunity to temperature-susceptible inbred lines?

Response: Thank you very much for this good question and suggestion, we performed several experiments to clarify this question. The first experiment was to study the expression patterns of *CaKAN3* and *CaHSF8* in diverse pepper inbred lines with different level of RSHT resistance, and found that the transcript levels of *CaKAN3* were much higher in lines with higher level of RSHT resistance compared to lines with lower RSHT resistance, indicating that transcript levels of *CaKAN3* were positively related to pepper immunity against RSI under HTHH in different pepper lines. *CaHSF8* was upregulated generally by HTHH, but there was no close relationship between the transcript level of *CaHSF8* and pepper immunity against RSHT. The second experiment was to silence *CaKAN3* or *CaHSF8* in pepper lines with different level of thermo-tolerant immunity, and found that the silencing of *CaKAN3* or *CaHSF8* reduced significantly the pepper immunity in pepper lines with high level of thermo-tolerant immunity, but did not produce clearly phenotypic effect in pepper lines susceptible to RSI under HTHH. The third experiment was to transiently overexpress *CaKAN3* or *CaHSF8* in pepper lines with lower levels of thermo-tolerant immunity, and found that the overexpression of *CaKAN3* significantly reduced propagation of the inoculated RS, but that of *CaHSF8* did not produce obvious phenotypic effect. All these data indicate that *CaKAN3* is specifically upregulated by and is crucial for pepper immunity against RSHT, while *CaHSF8* is upregulated by HTHH, and its role in pepper immunity against RSHT is *CaKAN3* dependent. We have added these data in the result section and modified the Discussion section correspondingly, please see Figure S9, line 192 to line 209 in the result section and line 404 to line 407 in the Discussion section in the revised MS.

4. In Supplementary Data Fig.4b and 5j, 35S::CaKAN3-GFP and 35S::CaHSF8-GFP

transgenic *Nicotiana benthamiana* plants are more tolerant to *Ralstonia* infection at high temperatures. Does *Nicotiana benthamiana* plant have CaKAN3 and CaHSF8 homologs in its genome? Are NLR genes regulated by CaKAN3 protein conserved in *Nicotiana benthamiana*?

Response: Thank you very much for this good question, we have checked the genome of *Nicotiana benthamiana* and found that there are orthologs of CaKAN3 and CaHSF8 in *Nicotiana benthamiana* (Fig S14a). In addition, 5 NLR genes with high sequence similarities to five of the six NLR genes in pepper genome were found in *Nicotiana benthamiana* genome, and these five NLR genes were found to be targeted and activated by ectopic expressed CaKAN3 or CaHSF8 (Fig S14). Furthermore, the result from VIGS experiment showed that the silencing of three NLR genes in *Nicotiana benthamiana* reduced significantly *Nicotiana benthamiana* immunity against RSI under HTHH (Fig S17). All these results showed that ectopic overexpression of CaKAN3 or CaHSF8 might enhanced *Nicotiana benthamiana* immunity against RSI under HTHH at least partially by activating NLR genes. We added these data in the result section and modified the Discussion section correspondingly, please see line 272 to line 283 and line 306 to line 308 in the result section and line 427 to line 431 in the Discussion section in the revised MS.

5. Is thermo-tolerant immunity through CaKAN3 and CaHSF8 affect basal immunity. In Fig. 1e and Supplementary Data Fig. 5g, flg22-mediated ROS was compromised in knock-down plants of CaKAN3 and CaHSF8 genes. Can you test the phenotypes of CaKAN3 or CaHSF8 knock-downed plants against other virulent *Ralstonia solanacearum* stains?

Response: Thank you very much for this good question, to answer this question, we have performed experiment to study the role of CaKAN3 and CaHSF8 in pepper immunity against other two virulent RS strains and *Pst DC3000*, the result showed that the silencing of CaKAN3 or CaHSF8 increased significantly pepper susceptibility to infection of each of the two virulent RS strains as well as to the infection of DC3000, supporting that CaKAN3 and CaHSF8 act positively in basal immunity. We have added these data in the Result section and modified Discussion section correspondingly, please see Figure S10 and line 210 to line 218 in the result section and line 432 to line 434 in the Discussion section in the revised MS.

6. There are two different size bars on imaging data in Fig.2c, Supplementary Data Fig.3, and Supplementary Data Fig.5b. Also, the magnification of each image varies. Please use equal and consistent imaging conditions for a better comparison.

Response: Thank you very much for pointing out this error, we have adjusted the bars in the three figures. Please see the revised figures (Fig. S3, Fig. S6b and Fig. 2c) in the revised MS.

7. Please include control experiment results showing the expression of proteins in BiFC and split Luciferase experiments (Fig2c, Fig 5a).

Response: Thank you very much for this good suggestion, we have added data of immunoblotting to present the success of the proteins in BiFC and Luciferase experiments. Please see the corresponding revised figures (Fig. S5 and Fig. S21) in the revised MS.

Minor points

1. The image used in Supplemental Data Figure 5 is too small.

Response: thank you very much for this good suggestion, we have revised the original Figure S5 into three new figures, and the revised figures are not too small now, please see the revised figures (Fig. S6, Fig. S67 and Fig. S8) in the revised MS.

2. Typos need to be fixed (line 155)

Response: Thank you very much, we have revised this error. Please see the revised line 175 in the revised MS.

3. Please replot ChIP-qPCR results in Fig3e and Fig4d to show the relative abundance of IP:HA samples against IP:IgG controls.

Response: Thank you very much for this good suggestion, we have re-plotted the ChIP-qPCR results in Figure 3e and Figure 4d showing the relative abundance of IP:HA sample against the IP:IgG control. Please see the revised figures in the revised MS.

4. The font size is too small in several figures.

Response: Thank you very much for pointing out this error, we have increased the font size in all of the figures. Please see the corresponding figures in the revised MS.

5. Please provide uncropped gel/immunoblotting images.

Response: Thank you very much for this good suggestion, we have replaced the images with the original ones, please see the following crude photographs.

Fig.2e

Fig.2f

Fig.2g

Fig.2h

Fig.S4a

Fig.S5

Fig.S8a

Fig.S9f

Fig.S21

Fig.5d

Fig.5e

6. Please spell out abbreviations (line 112, 134, 137, 139, 172).

Response: Thank you very much for this good suggestion, and have spelled out all of the abbreviations the lines you listed. Please see the revised line 127-128, line 151, line 155-156, line 158 and line 220-221 in the revised MS.

Reviewer #2 (Remarks to the Author):

Dear Editor/Authors,

Yang and co-authors have conducted a detailed and comprehensive investigation into the roles of pepper KAN3 and HSF8 (and their interaction) in temperature-specific disease resistance and thermotolerance. The authors showed that heat-induced KAN3 is a positive regulator of pepper immunity specifically at high temperature (37C) but not at the control temperature (28C). Additionally, they demonstrated that KAN3 interacts with HSF8 in vitro and in vivo, while also dissecting the KAN3 and HSF8 protein domains mediating this interaction. In terms of transcriptional regulation, KAN3 and HSF8 ChIP-Seq and EMSA showed enriched recruitment to immunity-associated intracellular immune receptor (NLR) genes, which exhibit enhanced expression at 37C. Mechanistically, loss of KAN3 reduces HSF8 binding to target NLR promoters and vice-versa. The authors then attempt to reveal a novel context-specific mechanism, wherein higher heat shock temperatures (42C) reduce NLR gene expression but induce HSP gene expression via thermosensitive recruitment of HSF8.

This study is quite novel and exciting as it presents detailed characterization of agriculturally important solanaceous plants using both VIGS and overexpression analyses. Comprehensive (and complementary) assays are also performed at the molecular, biochemical and physiological levels. Overall, the research presented in this manuscript should be significant to a broad readership, since it is at the nexus of gene regulation, plant immunity and plant stress biology.

To improve the manuscript, some issues need to be addressed and/or clarified:

1) The central conclusion of the paper is the differential interaction between KAN3 and HSF8 with each other and with target defence promoters under different temperatures. This is shown in the authors' model in Figure 5g. However, direct evidence on the main thesis of the paper is a bit lacking in my opinion. Can the authors directly demonstrate that KAN3-HSF8 protein interaction is differentially regulated by temperature using BiFC at 28C vs. 37C vs. 42C/45C.

Response: Thank you very much for this good suggestion, we have performed BiFC experiment to study CaKAN3-CaHSF8 interaction under different temperature conditions, the result was consistent to the results from CoIP and Luc assay that CaKAN3 interacted with CaHSF8 at 37°C but not at 28°C or 42°C. We added this result in the result section, please see Fig. 5a and Line 367 to Line 375 to Line 378 in the revised MS.

2) Although the authors show HSF8 ChIP analyses of target NLR and HSP genes at different temperatures (Supplementary Figure 14), the corresponding KAN3 ChIP

under different temperatures (28C, 37C and 42C) seems to be missing. Supplementary Fig 7 only shows KAN3 promoter binding dataset at one temperature. To fully support the authors conclusions and claims, I believe that this matter needs to be clarified.

Response: Thank you very much for this good suggestion, we have performed an experiment to study whether CaKAN3 target the promoters of the tested NLR genes using ChIP-qPCR under different temperature, result showed that the binding of CaKAN3 to the promoters of the teste NLR genes at 28 °C, 37 °C and 45°C, and these bindings were not affected significantly by the temperature. As CaKAN3 can not target HSP genes such as CaHSP17.4B, CaHSP18.2, CaHSP70 and CaHSP70-15 that were bound by CaHSF8, so we did not detect the effect of temperature on the binding of the HSPs by CaKAN3. We have added this data in the result section, please see Fig. S23c-d and Line 390 to Line 392 in the revise MS.

3) The authors conclude that temperature-regulated HSF8 recruitment leads to thermosensitive NLR gene expression. However, there is no difference in HSF8 recruitment to 4 NLR genes (CaR1B23, CaR1B11, CaR1A, CaR1B12) between 28C and 37C (Supplementary Figure 14b). Differential HSF8 recruitment at 28C vs. 37C is only observed in two NLR genes CaR1A6 and CaR1B16. How do the authors reconcile these with their data showing upregulated expression of all 6 NLR genes at 37C (Figure 4a-b)? I believe that this needs to be properly discussed in the paper.

Response: Thank you very for this good question, yes, we did find that there was no difference between CaHSF8 recruitment to the majority of the tested NLR genes between 28 °C and 37 °C, but significant difference was found in the transcription of these NLR genes between 28 °C and 37 °C, this inconsistency might be due to that both CaKAN3 and CaHSF8 are upregulated by high temperature induction (Fig S2b and S6a), 37 °C activates the expression of CaKAN3 and CaHSF8 compared with 28 °C, but the transcriptional activation of CaKAN3 was limited after 1 hpt of 37 °C, which may be the reason for this result. We mainly focus on 28 °C and 37 °C compared with 45 °C, because at 45 °C, CaHSF8 is present, but cannot regulate NLR transcription.

4) The Discussion section needs to synthesize the authors' comprehensive findings and then properly relate it to the broader and relevant literature. How do KAN3 and HSF8 directly connect to the cytokinin-mediated immunity required at high temperature that the authors previously discovered (Yang et al., 2021 Plant, Cell & Environment)? Do KAN3 and HSF8 directly regulate Mgst3 and PRP1 by binding those gene promoters as well?

Response: Thank you very much for this good suggestion, we have performed a ChIP-qPCR assay and found that neither Mgst3 nor PRP1 was directly targeted by KAN3 and HSF8. Based on the results in the present study and that from our previous study (Fig. S13 and line 266-269), we added a paragraph in the Discussion Section to describe how do KAN3 and HSF8 connect to the cytokinin-mediated immunity required at high temperatures, this paragraph is " It is worth pointing out that although cytokinin signalling has been implicated in plant immunity and in the trade off between plant immunity and growth/development^{21,100}, there is no report on its relationship to NLR and KANADI/HSF so far, we speculate that upon the challenge of combined high temperature and high humidity (31-37 °C, 90% humidity), CaKAN3/CaHSF8 is rapidly

activated and interacts with each from 1 to 3 hpt in a pathogen independent manner, which in turn activates NLRs. These NLRs might further activate high-temperature-high-humidity specific immunity in the presence of pathogen infection upon perception of pathogen ligands by activating CaMgst3 and CaPRP1 probably with action of cytokinin signalling initiated by pathogen infection under high temperature and high humidity conditions partially through chromatin activation¹⁵. If there is no pathogen infection after 3 hours of high temperature and high humidity treatment, or under extreme high temperatures (more than 42 °C), CaHSF8 no longer interacts with CaKAN3, and the released HSF8 alone activates HSP genes by directly binding the promoters and thus activates thermotolerance (Figure 5h). To elucidate the molecular mechanisms underlying functional association between cytokinin signalling and NLR, further study is required in the future.", please see Line 460 to Line 473 in the revise MS.

5) In relation to #4, why did KAN3 not activate transcription in the GAL4 Y2H? Is this somehow related to its putative role as a transcriptional repressor? If KAN3 belongs to a family of transcriptional repressors, how does this link with the current study showing transcriptional activation of NLR genes by KAN3? Does KAN3 switch between repressor and activator activities depending on temperature and/or protein interactors? Perhaps there could be a discussion of other transcription factors that exhibit this context-specific functional switch.

Response: Thank you very for this good question, we did not detect any transcriptional activity for KAN3 in yeast, and the transcription activity assay by GUS reporter gene consistently showed that CaKAN3 is a transcriptional repressor (Fig.S11c and Fig. 4a), it can be speculated that in the absence of HSF8, CaKAN3 acts as a transcriptional repressor, in this way KAN3 independently regulates 518 target gene by ChIP-seq (Fig. 3b). Despite that CaKAN3 act as a transcriptional repressor, it has no DLSL domain which is responsible for transcriptional repression, implying that the transcriptional activity of CaKAN3 might be determined by other proteins such as HSF8 through protein-protein interaction. As a matter of fact, we found that the presence of HSF8 transform KAN3 from transcriptional repressor to a transcriptional activator. We accordingly modified the discussion section, please see Line 446 to Line 451 in the revised MS.

6) Although the methodology is sound and comprehensive, I believe additional experimental details are needed to ensure future reproducibility by other researchers.

- Plant materials: Please indicate the supplier for the soil mix and the dimensions of the pots.

Response: Thank you very much for this good suggestion. We have added the supplier for the soil mix and the dimensions of the pots, please see in line 476-477 in the revised MS.

- Vector Construction: Please provide sources/citations for the plasmids. If applicable, indicate the AddGene number.

Response: Thank you very much for this good suggestion. We have added sources/citations for the plasmids and the supplier and item number of other plasmids, please see in line 485-487 in the revised MS.

- VIGS assay: How was GV3101 transformed with the VIGS or empty vector? Please

state or add a reference.

Response: Thank you very much for this good suggestion, we have described the method of agrobacterium transformation and added references, please see in line 496 in the revised MS.

- Bacterial treatments: Please detail how *R. solanacearum* strain Fjc100301 was cultured and prepared for infection assays. Also reference the source for the bacterial strain.

Response: Thank you very much for this good suggestion, we have described the detail of *R. solanacearum* culture and preparation for infection assays and have also added references, please see in line 502-507 in the revised MS.

- LC-MS/MS analysis: How was the IP portion of the IP-MS experiment performed? How was phosphopeptide enrichment performed? Was the “match between runs” feature used during the MaxQuant search? How was the data normalized for the proteomics analysis?

Response: Thank you very much for this good suggestion, the IP portion of the IP-MS experiment has been carried using the same method in CoIP experiment, we have provided details in the Method and Material Section. The "phosphopeptide" was error and we have corrected it, please see in line 527-539 in the revised MS. We used Proteome Discover 2.4. but not MaxQuant software to search the proteome database. We performed qualitative analysis rather than quantitative analysis and did not normalized the data in LC-MS/MS analysis.

- Subcellular localization and bimolecular fluorescence complementation (BiFC) assay: What were the specific components of the agroinfiltration buffer? How were GV3101 cells prepared (media and growth conditions)? What settings were used for the laser scanning confocal microscope in subcellular localization and BiFC experiments? For example, what were the excitation and emission settings? What time of scanning method was used and how long? What objective lens was used? What was the voxel size and area size for scanning?

Response: Thank you very much for this good suggestion, we have described the details of subcellular localization and BiFC assay, please see in line 555-561 and line 567-570 in the revised MS.

- Prokaryotic Expression: How was *E. coli* transformation conducted? How was SDS-PAGE of bacterial proteins done? What supplies/conditions were used? Was it the same as in the next subsection of your Methods? Please provide details.

Response: Thank you very much for this good suggestion, we have described the details of prokaryotic expression, please see in line 574-586 in the revised MS.

- Coimmunoprecipitation (Co-IP) and western blot analysis: What bacterial cells (Line 415) were used? Please be specific.

Response: Thank you very much for this good suggestion, we have described the details of CoIP assay, please see in line 589-590 in the revised MS.

- Genetic Transformation of *Nicotiana benthamiana*: Indicate the supplier of glufosinate used. Specify components, concentration and suppliers of the MS media.

Response: Thank you very much for this good suggestion, we have described the details of *Nicotiana benthamiana* genetic transformation, please see in line 622-625 in the revised MS.

- Chromatin immunoprecipitation (ChIP) and ChIP-seq: How was the fixation quenched (e.g. glycine? What concentration?)? Please include the brand and settings of the sonicator. How much antibody was used for ChIP? How was the ChIP DNA purified in detail? What software or R package was used to remove the low-quality reads and to trim the 3' linker sequences?

Response: Thank you very much for this good suggestion, we have described the details of ChIP assay, please see in line 632-640 in the revised MS.

- EMSA analysis: Please provide details in the Supplementary Information for easier reproducibility by other researchers.

Response: Thank you very much for this good suggestion, we have added the description for the experiments in EMSA, please see in line 662-669 in the revised MS.

- RT-qPCR assay: How was total RNA extracted? What type of QC was performed (A260/A280, A260,230, RIN scores, etc.)? What cutoff was used for subsequent gene expression analyses? How was cDNA synthesis performed? Please cite other studies that used CaActin as an internal control gene.

Response: Thank you very much for this good suggestion, we have described the details of RT-qPCR assay, since we did not use Agilent 2000 for RNA quality analysis, which is commonly used for high-throughput sequencing, we cannot provide a standard for RIN scores, please see in line 671-689 in the revised MS.

- DAB staining and chlorophyll fluorescence spectrophotometry; Provide details on how warm water treatment was performed on the plants. How old were these plants? Were they directly submerged individually or in plates?

Response: We apologize for this misrepresentation, We did not use warm water for high temperature and high humidity treatment. We have corrected and added more details for DAB staining and chlorophyll fluorescence spectrophotometry, please see in line 698-703 in the revised MS.

- ROS measurements: Indicate the supplier of flg22.

Response: Thank you very much for this good suggestion, we have added the supplier of flg22, please see Line 722 in the revised MS.

7) Reading through the text, the use of abbreviations can be toned down. I was a bit overwhelmed by the ubiquitous use of acronyms (RTHH, HTHH, RSRT, RSHT, ETHH, etc.). Would it be possible to rewrite the manuscript to minimize acronym usage and just state the actual temperature used? This would improve the article's readability.

Response: Thank you very much for this good suggestion, we tried to replace all of the acronyms with actual temperature and humidity, but in this way, the manuscript become very lengthy, which affects the readability of the paper. So, to make the MS more readable, we have adjusted the usage of the acronyms by replacing some of these acronyms, especially in the Results Section, with

actual temperature and humidity.

Figures and Tables:

- All figure captions (including supplementary figures): Description of the experimental design and statistical analyses should be further clarified. Please define what the sample n (biological replicates) means. Are they individual leaves from different plants? Are they pooled leaves from different plants? Are they different leaves but from the same plant?

Response: Thank you very much for this good suggestion, for the accuracy of the experimental results, all the replicates we used were from different leaves of different plants, and the numbers of the replicates were added to the associated Figures.

- All figure captions (including supplementary figures): For every panel, please indicate the exact number of experimental repeats conducted that led to reproducible results shown by the representative figure.

Response: Thank you very much for this good suggestion, we have added the exact number of experimental repeats.

- Figure 2e: the anti-His bands for the EV 6X-His setup is missing in the Input immunoblot. Please make sure to show them and avoid overcropping of the blot.

Response: Thank you very much for this good suggestion, probably due to transmembrane or antibodies we used or other unknown factors, no strips were exposed in the lane corresponding to EV, so we replace the EV with CaKAN4-6X-His. Please see Figure 2e in the revised MS.

- Figure 2f: The KAN4 IP shows pulldown of HSF8 but not the KAN3 IP. Was this switched because it is not consistent with the authors' conclusions indicating KAN3-HSF8 interaction?

Response: Thank you very much for your careful consideration, we have corrected this error, please see in Figure 2f in the revised MS.

- Figure 3f: The blots seem eerily similar even though they are separate experiments with different probes. To be safe, please recheck this figure for potential/inadvertent image duplications.

Response: We have carefully checked Figure 3f, the two photographs were similar but not duplicate one, we have repeated the experiment and some photographs were replaced with new photos, now the photographs are not so similar to each other, please see the Figure 3f in the revised MS.

- Supplementary Figure 2: The RTHH acronym is not defined in the caption.

Response: Thank you very much for your careful consideration, we have corrected this error, please see in Figure S2 in the revised MS.

- Supplementary Figure 4b: Please add the time course disease index data (similar to Supplementary Figure 5e)

Response: Thank you very much for this good suggestion, we have added the disease index data, please see in Figure S4 and Figure S8 in the revised MS.

- Supplementary Figure 4c: Please indicate which is RSRT and RSHT.

Response: Thank you very much for this good suggestion, we have added the disease index data, please see in Figure S4c in the revised MS.

- Supplementary Figure 5k: Please indicate which is RSRT and RSHT.

Response: Thank you very much for this good suggestion, we have added the disease index data, please see in Figure S8d in the revised MS.

- Supplementary Figure 6c: Because the authors are comparing between two temperatures, the statistical analyses should include the comparison between RTHH and HTHH. Also, the caption should state “yeast one-hybrid” not “yeast two-hybrid.”

Response: Thank you very much for this good suggestion, we have used LSD test again to compare the samples of each group, and replaced “yeast two-hybrid.” with “yeast one-hybrid”, please see in Figure S11c and d in the revised MS.

- Supplementary Figure 11: Quantify the survival rates and perform statistical analyses in Panel a. Also, the acronym HTS is not defined.

Response: Thank you very much for this good suggestion, we have added the survival rates in Fig S19a, and replaced "HTS" with "extreme high temperature treatment", please see in Fig S19a and Fig S19g in the revised MS.

- Supplementary Figure 14b: Statistical analyses of the graphs are missing.

Response: Thank you very much, we have performed statistical analyses using LSD for Figure S22b.

- Supplementary Table 1: Indicate the sources for all primers or probes (i.e. previous publication or this study?)

Response: Thank you very much for this good suggestion, we have added references to the primers used in previous studies, please see in Table S1 in the revised MS.

- Supplementary Table 2: Reference the research that first used this disease index. Or was this implemented for the first time in this study?

Response: Thank you very much for this good suggestion, we have added the references that first used this disease index, please see in Table S2 in the revised MS.

Text:

- Line 37; Make sure to cite your opening sentence, perhaps using articles that have examined at the impact of temperature and humidity on plant diseases. Generally, it would be good to expand the literature context in this paragraph. That is, I would recommend synthesizing what is currently known on how environmental factors influence plant

immunity and plant-pathogen interactions.

Response: Thank you very much for your good suggestion, we have re-organized the first paragraph in the Introduction Section following your suggestion, please see the Line 36 to 38 in the revised MS.

- Line 41: This sentence should have a citation.

Response: Thank you very much for your good suggestion, we have added reference at Line 41, please see in line 38-43 in the revised MS.

- Lines 46-61: Because your study looks at pepper disease resistance, I would recommend also citing literature on plant immune receptors and signaling specifically in solanaceous plants.

Response: Thank you very for your good suggestion, we have added references associating with immune receptors and signaling specifically in solanaceous plants. Please see Line 41 to Line 47 in the revised MS.

- Line 60-61: This phrase should be revised in the context of pepper plants: “how TFs function in plant immunity against pathogens under HTHH is unclear.” There have been some studies on transcription factors involved in temperature-regulated immunity in other plants (like Arabidopsis), e.g. CAMTA3, PIF4, CBP60g, SARD1.

Response: Thank you very much for your good suggestion, we have reorganized this sentence to “Accumulating evidence suggest that, except for through differentially modulating signaling mediated by phytohormones such as SA, JA and cytokinins^{4,15}, plant immunity can also be compromised by high temperature stress or high humidity through repressing NLR proteins such as SNC1, RPW8.1 and RPW8.27,37-40 and TFs such as CAMTA3, PIF4, CBP60g, SARD1⁴¹⁻⁴³, indicating that the modification of plant immunity by high temperature or high humidity might occur at multiple levels including pathogen effector perception and transcriptional level. However, the NLR proteins and TFs involved in high-temperature-high-humidity specific plant immunity and how they are functionally related remain to be elucidated.”.Please see Line 68 to Line 76 in the revised MS.

- Lines 93-99: This is one long sentence that could negatively affect readability. I would suggest breaking this up into a few shorter sentences.

Response: Thank you very much for your good suggestion, we have reorganized this long sentence as “In a dataset of RNA-seq using pepper roots challenged with RSI at 37 °C, 90% humidity, a gene encoding a putative KAN3 attracted our attention. Its deduced amino acid sequence contains a conserved GARP domain but does not contain the DLSL domain, and appear to be structurally conserved in the Capsicum genus, including Capsicum annum, Capsicum baccatum and Capsicum chinense (Supplementary Fig. 1a, 1b), and we named it CaKAN3. In addition, several cis-elements including AT-rich, W-box, ATCT, GARE- and G/C motifs were found within the promoter region of CaKAN3 (Supplementary Fig. 2a).”. Please see Line 108 to Line 114 in the revised MS.

- Line 103: I would suggest stating Nicotiana benthamiana instead of NB since the paper

already has lots of acronyms. This would enhance readability.

Response: Thank you very much, we have replaced all of the NBs with *Nicotiana benthamiana*.

- Lines 314-316: Please also discuss studies showing temperature regulation of solanaceous NLRs and ETI (e.g. in tomato, tobacco, etc.) since your study deals with Solanaceae.

Response: Thank you very much for this good suggestion, we have added the NLRs and ETI in solanaceous plants in the Discussion Section, please see Line 420 to Line 425 in the revised MS.

- Line 317: It would be good to relate this to studies showing more effective NLR-mediated ETI at high temperatures, e.g. Sr21, Sr13 in wheat and Xa7 in rice.

Response: Thank you very much for this good suggestion, we have added the NLRs that effectively mediate ETI at high temperatures in the Discussion Section, please see Line 425 to Line 426 in the revised MS.

- Line 486: Please define the acronym CCD.

Response: Thank you very much for your good suggestion, the CCD is Charge coupled Device, please see the Line 695 in the revised MS.

REVIEWER COMMENTS

Reviewer #1 (Remarks to the Author):

The authors have addressed most of my previous comments and the revised manuscript has improved. However, there are two remaining outstanding issues that need to be addressed editorially (no new experiments):

1. The abstract needs to be further modified. As I pointed out in my previous review, high temperature and high humidity are known to induce plant susceptibility to a variety of diseases including bacterial wilt studied here. Only in this broad context, CaKAN3-mediated NLR resistance PARTIALLY increases plant resistance to bacterial wilt. The abstract needs to clearly state this observation. I suggest the following modification of the first sentence to make it more accurate and understandable to general readers. "It was previously found that high temperature and high humidity (HRHH) conditions increase plant susceptibility to a variety of diseases, including bacterial wilt in solanaceous plants. However, SOME solanaceous plant cultivars have evolved a mechanism to activate high-temperature-specific immunity to cope with bacterial wilt disease. The underlying mechanisms remains poorly understood. Herein, we found that upon *R. solanacearum* inoculation (RSI)..."

2. In the results section, please describe Fig. 1c and 1d more accurately: "Consistent with previous studies, the HTHH condition SUPPRESSES plant resistance to bacterial wilt disease (Fig. 1c). Interestingly, however, CaKAN3 silencing FURTHER increases plant susceptibility to bacterial wilt under the HTHH condition, but not under RSRT conditkion (Fig. 1d)..."

Reviewer #2 (Remarks to the Author):

Dear Editor/Authors,

I would like to commend the authors for the herculean task in comprehensively revising this manuscript! My major concerns as a reviewer have now been addressed in this important paper. As mentioned previously, this work is quite exciting and novel. It has provided mechanistic insights into Solanaceae plant disease resistance in a changing climate, which I believe would be of significant interest to a broad readership.

A few suggestions for revisions:

1) The authors have now performed the crucial KAN3-HSF8 interaction experiments under different temperatures using BiFC in Figure 5a. Although the results support the authors' conclusions, the experiment seems to be lacking in controls. It would benefit the paper if this BiFC experiment is presented in the same detail as their BiFC datasets in Figure 2c.

2) The paper now shows KAN3 ChIP under different temperatures (28C, 37C and 45C) in Figure S23c, and the authors conclude that there is no significant effect. However, their statistical analyses show that KAN3 recruitment is lower at 45C for the CaR1B23, CaR1A and CaR1B16 promoters, but higher at 45C in the CaR1A6 promoter. These findings need to be properly acknowledged and mentioned in the Results and Discussion sections.

3) The differential HSF8 recruitment at 28C vs. 37C is only observed in two NLR genes CaR1A6 and CaR1B16 (Figure S23B). This is explained in the response letter to potentially be due to CaKAN3 upregulation at 37C being limited to just the 1-hour timepoint. I believe this interesting observation should at least be stated in the Results and/or Discussion sections.

4) The Discussion has been modified to mention that the presence of HSF8 potentially transforms KAN3 from a transcriptional repressor to a transcriptional activator. In my opinion, it would broaden the paper's significance and implications if there is a discussion of other transcription factors that exhibit this context-specific functional switch.

5) In Figure 2f: The co-IP experiment actually shows interaction between HSFA1 and KAN3 (first lane of the IP: anti-Myc blot) but not HSF8 and KAN3 (second lane of the IP: anti-Myc blot). I am a bit confused by this data – can the authors clarify this?

6) The authors present data that the 6 NLRs could potentially form one functional complex but I do not think that there is sufficient data to conclude that these form a resistosome complex (Line 437). I would recommend toning down the language in this part of the Discussion.

7) I would suggest adding an overarching concluding statement in the final paragraph of the Discussion section.

8) Please indicate the temperature used in the caption for Figure 4c.

Comments on the text:

- Lines 19-21: Can the first sentence of the abstract be rephrased for better clarity?
- Lines 45-47: I would suggest re-stating this sentence, since I was a bit confused.
- Lines 68-74: I believe that this sentence can be improved for better readability.
- Lines 420-425: I would suggest re-stating this sentence to improve its readability.
- Line 422: Just to clarify: Cf-4 and Cf9 are R proteins but they are not NLRs (they are cell-surface receptor proteins).
- Lines 425-432: Please break up into more concise sentences to increase the clarity of the text.

REVIEWER COMMENTS

Reviewer #1 (Remarks to the Author):

The authors have addressed most of my previous comments and the revised manuscript has improved. However, there are two remaining outstanding issues that need to be addressed editorially (no new experiments):

1. The abstract needs to be further modified. As I pointed out in my previous review, high temperature and high humidity are known to induce plant susceptibility to a variety of diseases including bacterial wilt studied here. Only in this broad context, CaKAN3-mediated NLR resistance PARTIALLY increases plant resistance to bacterial wilt. The abstract needs to clearly state this observation. I suggest the following modification of the first sentence to make it more accurate and understandable to general readers. "It was previously found that high temperature and high humidity (HRHH) conditions increase plant susceptibility to a variety of diseases, including bacterial wilt in solanaceous plants. However, SOME solanaceous plant cultivars have evolved a mechanism to activate high-temperature-specific immunity to cope with bacterial wilt disease. The underlying mechanisms remains poorly understood. Herein, we found that upon *R. solanacearum* inoculation (RSI)..."

Response: Thank you very much for your kindness and good suggestion, we have replaced the original sentence with your excellent one, please see Line 19 to Line 22 in the revised MS.

2. In the results section, please describe Fig. 1c and 1d more accurately: "Consistent with previous studies, the HTHH condition SUPPRESSES plant resistance to bacterial wilt disease (Fig. 1c). Interestingly, however, CaKAN3 silencing FURTHER increases plant susceptibility to bacterial wilt under the HTHH condition, but not under RSRT condition (Fig. 1d)..."

Response: Thank you again for your kindness and good suggestion, we have replaced the original sentence with your excellent sentence, please see Line 122 to Line 124 in the revised MS.

Reviewer #2 (Remarks to the Author):

Dear Editor/Authors,

I would like to commend the authors for the herculean task in comprehensively revising this manuscript! My major concerns as a reviewer have now been addressed in this important paper. As mentioned previously, this work is quite exciting and novel. It has provided mechanistic insights into Solanaceae plant disease resistance in a changing climate, which I believe would be of significant interest to a broad readership.

A few suggestions for revisions:

1) The authors have now performed the crucial KAN3-HSF8 interaction experiments

under different temperatures using BiFC in Figure 5a. Although the results support the authors' conclusions, the experiment seems to be lacking in controls. It would benefit the paper if this BiFC experiment is presented in the same detail as their BiFC datasets in Figure 2c.

Response: Thank you very much for your good suggestion, we have performed another BiFC experiment with necessary controls, because the photograph is too big, we combine these data with data to confirm the success of transient overexpression of CaHSFA1-GLUC, CaHSF8-CLUC, CaKAN3-NLUC and CaKAN4-NLUC, which were listed in Supplementary Data Fig.21a.

2) The paper now shows KAN3 ChIP under different temperatures (28C, 37C and 45C) in Figure S23c, and the authors conclude that there is no significant effect. However, their statistical analyses show that KAN3 recruitment is lower at 45C for the CaR1B23, CaR1A and CaR1B16 promoters, but higher at 45C in the CaR1A6 promoter. These findings need to be properly acknowledged and mentioned in the Results and Discussion sections.

Response: Thank you very much for this good suggestion, we have rewritten the sentence in "CaKAN3 bound the promoters of 6 tested NLR genes at 28 °C, 90% humidity, 37 °C, 90% humidity and 45 °C, 90% humidity, and no significant difference was found between the treatments and two time points (1 and 6 hpt) (Supplementary Fig.23 c and d)." in the result section as "CaKAN3 bound the promoters of 6 tested NLR genes at 28 °C, 90% humidity, 37 °C, 90% humidity and 45 °C, 90% humidity with slightly different affinities under the three tested conditions, but no significant difference was found between the treatments and two time points (1 and 6 hpt) (Supplementary Fig.23 c and d)." We have also rewritten the sentence in the Discussion section as "On the other hand, there was only slight difference in the bindings of CaKAN3 to the tested NLR promoters between 28 °C and 37 °C, but this difference was not regular and was not affected by CaHSF8 (Supplementary Fig.23)." Please see Line 447 to Line 449 in the Discussion Section.

3) The differential HSF8 recruitment at 28C vs. 37C is only observed in two NLR genes CaR1A6 and CaR1B16 (Figure S23B). This is explained in the response letter to potentially be due to CaKAN3 upregulation at 37C being limited to just the 1-hour timepoint. I believe this interesting observation should at least be stated in the Results and/or Discussion sections.

Response: Thank you very much for your good suggestion, we have rewritten a sentence "Furthermore, we found that CaHSF8 bound the promoters of the six NLRs with slightly difference in their binding affinities at 28 °C, 90% humidity and 37 °C, 90% humidity, but did not bind the promoters of NLR genes at 45 °C, 90% humidity(Supplementary Fig.23a,b). " in the result section. We also mentioned this difference in the Discussion section as "It is worth pointing out that compared with that at 28 °C, the binding of CaHSF8 to NLRs promoters was slightly enhanced at 37 °C, which may be attributed to its enhanced recruitment to the tested NLR genes by CaKAN3 that was upregulated at 37 °C." Please see Line 445 to Line 447 in the Discussion Section.

4) The Discussion has been modified to mention that the presence of HSF8 potentially transforms KAN3 from a transcriptional repressor to a transcriptional activator. In my opinion, it would broaden the paper's significance and implications if there is a discussion of other transcription factors that exhibit this context-specific functional switch.

Response: Thank you very much, we have added a reference about that “WRKY70 was previously found to be turned from a negative regulator for SARD1 expression to a positive regulator by phosphorylation to support our speculation”. Please see Line 440 to Line 442 in the Discussion Section.

5) In Figure 2f: The co-IP experiment actually shows interaction between HSFA1 and KAN3 (first lane of the IP: anti-Myc blot) but not HSF8 and KAN3 (second lane of the IP: anti-Myc blot). I am a bit confused by this data – can the authors clarify this?

Response: Thank you very much for pointing this error, we have corrected this mistake. Please see Fig.2f in the revised MS.

6) The authors present data that the 6 NLRs could potentially form one functional complex but I do not think that there is sufficient data to conclude that these form a resistosome complex (Line 437). I would recommend toning down the language in this part of the Discussion.

Response: Thank you very much for this good suggestion, we have rewritten this sentence as “The synergistic relationship among the NLR proteins was further supported by the data that they probably form a resistosome like complex”. Please see Line 429 in the revised MS.

7) I would suggest adding an overarching concluding statement in the final paragraph of the Discussion section.

Response: Thank you very much, we have added a concluding statement following your suggestion. Please see Line 473 to Line 478 in the revised MS.

8) Please indicate the temperature used in the caption for Figure 4c.

Response: Thank you very much, we have indicated the temperature used in Figure 4c.

Comments on the text:

- Lines 19-21: Can the first sentence of the abstract be rephrased for better clarity?

Response: Thank you very much, we have rewritten the first sentence of the abstract, please see Line 19 to Line 22 in the revised MS.

- Lines 45-47: I would suggest re-stating this sentence, since I was a bit confused.

Response: Thank you very much, we have rewritten this sentence as “As a matter of fact, plants have evolved high-temperature-high-humidity specific immunity to compensate the immunity impaired by conditions of high-temperature-high-humidity”, Please see Line 45-47 in the revised MS.

- Lines 68-74: I believe that this sentence can be improved for better readability.

Response: Thank you very much for this good suggestion, we have rewritten this part as “Accumulating evidence suggest that high temperature or high humidity can compromise plant immunity not only by inhibiting SA, JA or cytokinins signaling, but also through repressing NLR proteins such as SNC1, RPW8.1 and RPW8.2 and TFs such as CAMTA3, PIF4, CBP60g, SARD1, indicating that the modification of plant immunity by high temperature or high humidity might occur at multiple levels including pathogen effector perception and transcriptional level.”, please see Line 67 to Line 72 in the revised MS.

- Lines 420-425: I would suggest re-stating this sentence to improve its readability.

Response: Thank you very much for this good suggestion, we have rewritten this part as “Effector-triggered immunities have frequently been found to be repressed by high temperature or high humidity through repressing NLRs. Despite that fact that NLR proteins in solanaceous plants such as Tsw in pepper, Bs4, SISR-1 and I2 in tomato, ZAR1 and Roq1 in Nicotiana benthamiana have been functionally characterized in ETI, the effect of high temperature or high humidity on NLR mediated immunity in solanaceous plants is currently unclear.” Please see Line 412 to 416 in the revised MS.

- Line 422: Just to clarify: Cf-4 and Cf9 are R proteins but they are not NLRs (they are cell-surface receptor proteins).

Response: Thank you very much for pointing out this mistake, we have deleted this, please see Please see Line 414 in the revised MS.

- Lines 425-432: Please break up into more concise sentences to increase the clarity of the text.

Response: Thank you very much for this good suggestion, we have rewritten this paragraph following your suggestion, please see Line 416 to 424 in the revised MS.

REVIEWERS' COMMENTS

Reviewer #2 (Remarks to the Author):

Dear Authors/Editor,

Yang and co-authors have addressed all my concerns as a reviewer. Thank you very much for this revised submission and for reporting a comprehensive, timely and insightful study.

REVIEWERS' COMMENTS

Reviewer #2 (Remarks to the Author):

Dear Authors/Editor,

Yang and co-authors have addressed all my concerns as a reviewer. Thank you very much for this revised submission and for reporting a comprehensive, timely and insightful study.

Response: Thank you for your professional and meticulous comments, which is of great help to the improvement of this manuscript. This peer review has also brought us a lot of improvement. Thank you again!